



Mixing state of oxalic acid containing particles in the rural area of Pearl
River Delta, China: implication for seasonal formation mechanism of
Secondary Organic Aerosol (SOA)
Chunlei Cheng[1,2], Mei Li[1,2*], Chak K. Chan[3], Haijie Tong[4], Changhong Chen[5],
Duohong Chen[6], Dui Wu[1,2], Lei Li[1,2], Peng Cheng[1,2], Wei Gao[1,2], Zhengxu Huang[1,2],
Xue Li[1,2], Zhong Fu[7], Yanru Bi[7], Zhen Zhou[1,2*]
[1]Institute of Mass Spectrometer and Atmospheric Environment, Jinan University,
Guangzhou 510632, China
[2]Guangdong Provincial Engineering Research Center for on-line source apportionmen
t system of air pollution, Guangzhou 510632, China
[3]School of Energy and Environment, City University of Hong Kong, Hong Kong,
China
[4]Max Planck Institute for Chemistry, Multiphase Chemistry Department,
Hahn-Meitner-Weg 1, 55128 Mainz, Germany
[5]State of Environmental Protection Key Laboratory of the formation and prevention of
urban air pollution complex, Shanghai Academy of Environmental Sciences, Shanghai
200233, China
[6] State Environmental Protection Key Laboratory of Regional Air Quality Monitoring,
Guangdong Environmental Monitoring Center, Guangzhou, 510308, China
[7]Guangzhou Hexin Analytical Instrument Limited Company, Guangzhou 510530,
China
*Correspondence to*: Mei Li (limei2007@163.com) and Zhen Zhou (zhouzhen@gig.ac.cn)
Tel: 86-20-85225991, Fax: 86-20-85225991













**Abstract**:

The formation of oxalic acid and its mixing state in atmospheric particulate

matter (PM) were studied using a single particle aerosol mass spectrometer (SPAMS)
in the summer and winter of 2014 in Heshan, a supersite in the rural area of the Pearl
River Delta (PRD) region in China. Oxalic acid-containing particles accounted for 2.5%
and 2.7% in total detected ambient particles in summer and winter, respectively.
Oxalic acid was measured in particles classified as elemental carbon (EC), organic
carbon (OC), elemental and organic carbon (ECOC), biomass burning (BB), heavy
metal (HM), secondary (Sec), sodium-potassium (NaK) and dust. Oxalic acid was
found predominantly mixing with sulfate and nitrate during the whole sampling
period, likely due to aqueous phase reactions. In summer, oxalic acid-containing
particle number and ozone concentration followed a very similar trend, which may
reflect the significant contribution of photochemical reactions to oxalic acid formation.
Furthermore, favorable in-situ pH (2-4) conditions were observed, which promote
Fenton like reactions for efficient production of •OH in HM type particles. A
mechanism in which products of photochemical oxidation of VOCs partitioned into
the aqueous phase of HM particles, followed by multistep oxidation of •OH through
Fenton like reactions to form oxalic acid is proposed. In wintertime, carbonaceous
type particles contained a substantial amount of oxalic acid as well as abundant
carbon clusters and biomass burning markers. The general existence of nitric acid in
oxalic acid-containing particles indicates an acidic environment during the formation
process of oxalic acid. Organosulfate-containing particles well correlated with oxalic
acid-containing particles during the episode, which suggests the formation of oxalic
acid is closely associated with acid-catalyzed reactions of organic precursors.

**Keywords**: Oxalic acid; Single particles; Mixing state; Photochemical process;
Secondary organic aerosols.




## 1. Introduction

Organic aerosol, typically a large fraction of fine particles, contains more than thousands of organic compounds and contributes to visibility reduction, photochemical smog, climate change and adverse health effects (Novakov and Penner, 1993;Goldstein and Galbally, 2007;Jimenez et al., 2009;Poschl and Shiraiwa, 2015). A significant component of organic aerosol is secondary organic aerosol (SOA) formed from the gas phase oxidation of volatile organic compounds (VOCs) followed by partitioning of products into particles or from heterogeneous reactions of VOCs with particles (Hallquist et al., 2009;Zhang et al., 2015). Dicarboxylic acids (DCAs) are abundant and ubiquitous constituents in SOA and can be effective tracers for the oxidative processes leading to the formation of SOA (Kawamura and Ikushima, 1993;Ervens et al., 2011;Wang et al., 2012;Cheng et al., 2013). DCAs normally have high water solubility and low vapor pressure, so they play important roles in controlling the hygroscopic properties of organic aerosols (Prenni et al., 2003;Ma et al., 2013) and activating cloud condensation nuclei (Booth et al., 2009). The primary emissions of DCAs from anthropogenic sources are minor (Huang and Yu, 2007;Stone et al., 2010), and they are mainly derived from secondary oxidation of VOCs and subsequent intermediates (Ho et al., 2010;Myriokefalitakis et al., 2011). Even though high concentrations of DCAs have been observed in air masses influenced by biomass burning (Kundu et al., 2010;Kawamura et al., 2013), the primary source of DCAs is still not clear (van Pinxteren et al., 2014).

The production of DCAs through photochemical reactions has been reported in many field studies via the analysis of the diurnal and seasonal variations of DCA(Kawamura and Ikushima, 1993;Kawamura and Yasui, 2005;Aggarwal and Kawamura, 2008;Pavuluri et al., 2010;Ho et al., 2011), but the mechanism of DCAs formation is still not well understood. Oxalic acid is usually the most abundant DCA observed in the field (Kawamura et al., 2004;Ho et al., 2007;Kawamura et al., 2010). Several studies have found a tight correlation between oxalic acid and sulfate in ambient particles, implying that aqueous chemistry leads to the formation of oxalic





acid in aerosols and cloud droplets (Yao et al., 2002;Yao et al., 2003;Yu et al.,
2005;Sorooshian et al., 2007;Miyazaki et al., 2009). In recent years, many model and
laboratory studies suggest that the aqueous phase oxidation of highly water-soluble
organics like glyoxal, methylglyoxal and glyoxylic acid can efficiently produce oxalic
acid in aerosol particles and cloud droplets (Lim et al., 2010;Myriokefalitakis et al.,
2011;Ervens et al., 2014;Yu et al., 2014;McNeill, 2015). Recent stable carbon isotope
studies and field observations have also suggested that oxalic acid forms through
aqueous phase reactions (Wang et al., 2012;Cheng et al., 2015). However, the exact
formation pathways of oxalic acid in ambient particles are still unknown due to the
complexity of meteorological condition and the temporal resolution limitations of
conventional filter sampling studies and bulk chemical analysis.
Online measurements of the size distribution of oxalic acid-containing particles
and the mixing state of oxalic acid with other compounds in aerosols are useful to
examine the formation and evolution of oxalic acid and SOA particles. Sullivan and
Prather investigated the diurnal cycle and mixing state of DCA-containing particles in
Asian aerosol outflow using aerosol time-of-flight mass spectrometry (ATOFMS), and
proposed the formation of DCA on Asian dust (Sullivan and Prather, 2007). In
addition, Yang et al. (2009) measured oxalic acid particles in Shanghai and proposed
that in-cloud processes and heterogeneous reactions on hydrated aerosols contributed
to the formation of oxalic acid (Yang et al., 2009). So far the formation mechanism of
oxalic acid especially in urban areas is still not clear. Online measurements of the
mixing state of oxalic acid provides a powerful context to better understand the
formation of oxalic acid in aerosol particles and cloud droplets.
The Pearl River Delta (PRD) region has distinct meteorological seasonality
under the influence of the Asian monsoon system, which brings air from the ocean in
spring and summer, and carries polluted air from northern China in autumn and winter.
Strong photochemical activity occurs in summer under the condition of high
temperature and relative humidity, and in winter high loadings of particles from
northern cities are favorable for the occurrence of haze episode (Bi et al., 2011;Zhang
et al., 2013;Zhang et al., 2014). Here we present the seasonal field measurements of



the mixing state of oxalic acid-containing particles using a single particle aerosol
mass spectrometer (SPAMS) in a rural supersite of the PRD region. The seasonal
characteristic of oxalic acid particles and mixing state with secondary species were
investigated to explore the formation mechanisms of oxalic acid and aging process of
SOA.

## 135   2. Methods

### 136   2.1 Aerosol sampling

Particles were sampled using a single particle aerosol mass spectrometer
(SPAMS) at the Guangdong Atmospheric Supersite (22.73N, 112.93E), a rural site at
Heshan city (Figure S1). The supersite is surrounded by farm land and villages, with
no local industrial or traffic emissions. Ambient aerosols were sampled to the SPAMS
through a 2.5m long copper tube with 0.5m of the sampling inlet located above the top
of the building. The measurement period was from July 18 to August 1 in 2014, and
from January 27 to February 8 in 2015. Real-time $PM_{2.5}$ mass concentration was
simultaneously measured by a TEOM monitor (series 1405, Thermo scientific), and
hourly concentrations of $O_3$ were measured by an $O_3$ analyzer (model 49i, Thermo
scientific). The local meteorological data including temperature, relative humidity and
visibility were measured on the rooftop of the building. The average temperature
during the field study was 29.5 ℃ in summer and 14.1 ℃ in winter and the average
relative humidity was 71.7% and 63% in summer and winter, respectively.

### 150   2.2 SPAMS

Real-time measurements of single atmospheric particles has been demonstrated
by Prather and co-workers in the 1990s using aerosol time-of-flight mass
spectrometry (ATOFMS) (Prather et al., 1994;Noble and Prather, 1996). Based on the
same principle, the single particle aerosol mass spectrometer (SPAMS) developed by
Guangzhou Hexin Analytical Company was applied to field measurements of single
particles in the current work. The details of the SPAMS system have been introduced
previously (Li et al., 2011). Briefly, aerosol particles are sampled into the vacuum
pumped aerodynamic lens of the SPAMS through an electro-spark machined 80μm



critical orifice at a flow rate of 75 ml min$^{-1}$. The individual particles with a terminal
velocity are introduced to the sizing region. The velocity of each single particle is
detected by two continuous laser beams (diode Nd:YAG, 532 nm) with a space of 6
cm. The velocity is then used to calculate the single particle aerodynamic diameter
and provide the precise timing of the firing of a 266 nm laser used to induce
desorption and ionization (Nd:YAG laser, 266nm). The energy of the
desorption/ionization 266 nm laser was 0.6 mJ and the power density was kept at
about $1.6 \times 10^8$ W/cm$^2$ during both sampling periods. The 266 nm laser generates
positive and negative ions that are detected by a Z-shaped bipolar time of flight mass
spectrometer. The size range of the detected single particles is 0.2 to 2 μm.
Polystyrene latex spheres (Nanosphere size standards, Duke Scientific Corp., Palo
Alto) of 0.22-2.0 μm diameter were used for size calibration.

**171    2.3 Data analysis**

The size and chemical composition of single particles detected by SPAMS were

analyzed using the COCO toolkit based on the Matlab software. Particles were
clustered into several groups using the neural network algorithm (ART-2a) to group
particles into clusters with similar mass spectrum features. The ART-2a parameters
used in this work were set to a vigilance factor of 0.8, a learning rate of 0.05, and a
maximum of 20 iterations. We collected 516,679 and 767,986 particles with both
positive and negative mass spectra in summer and winter respectively. A standard
solution of oxalic acid was prepared with pure oxalic acid ($H_2C_2O_4$, purity: 99.99%,
Aladdin Industrial Corporation) and atomized to aerosols. After drying through two
silica gel diffusion driers, pure oxalic acid particles were directly introducing into the
SPAMS. The positive and negative mass spectra of oxalic acid are shown in Figure S2.
Based on the mass spectra of pure oxalic acid and previous ambient measurements by
ATOFMS (Silva and Prather, 2000;Sullivan and Prather, 2007;Yang et al., 2009),
$HC_2O_4^-$ (m/z -89) is selected as the ion peak for oxalic acid containing particles. In
this work, oxalic acid particles are identified if the peak area of m/z -89 was larger
than 0.5% of the total signal in the mass spectrum. With this threshold, 13109 and
20504 of oxalic acid-containing particles were obtained in summer and winter





separately, accounting for 2.5% and 2.7% of the total detected particles. According to
characteristic ion markers and dominant chemical species (Table S1), all oxalic acid
particles were classified into eight types: elemental carbon (EC), organic carbon (OC),
elemental and organic carbon (ECOC), biomass burning (BB), heavy metal (HM),
secondary (Sec), sodium-potassium (NaK) and dust.
**2.4 Inorganic ions and in-situ pH ($pH_{is}$)**
Water-soluble inorganic ions and trace gases were determined by an online
analyzer for monitoring aerosols and gases (MARGA, model ADI 2080, Applikon
Analytical B. V. Corp., the Netherlands) with a $PM_{2.5}$ sampling inlet at one hour
resolution from July 18 to August 1 in 2014. The principle and instrumental design
has been described in detail elsewhere (ten Brink et al., 2007;Du et al., 2011;Behera et
al., 2013;Khezri et al., 2013). Standard solutions containing all detected ions were
injected into MARGA before and after the field measurement. The liquid water
content and the concentration of $H^+$ in particles are calculated using the ISORROPIA
II model (Nenes et al., 1998, 1999;Fountoukis and Nenes, 2007). The in-situ pH ($pH_{is}$)
of particles is calculated through the following equation:
$$pH_{is} = -\log\alpha_{H^+} = -\log(\gamma_{H^+} \times n_{H^+} \times 1000/V_a) \qquad (1)$$
where $n_{H^+}$ is the concentration of $H^+$ (mol m$^{-3}$) and $V_a$ is the volume concentration of
the $H_2O$ (cm$^3$ m$^{-3}$) , while $\gamma_{H^+}$ is the activity coefficient of $H^+$(Xue et al., 2011;Cheng
et al., 2015). The temporal variation of $pH_{is}$ of ambient $PM_{2.5}$ particles is presented in
Figure S3, and demonstrated that 97% of particles were acidic in summer.
**3. Results and Discussion**
**3.1 Seasonal variation of oxalic acid containing particles**
The clustered 48 hr back trajectories of air masses arriving in Heshan during the
sampling period are shown in Figure S4. In summer, air masses at 500m levels above
the ground were mainly from the ocean and rural areas with less influence of human
activity, while in winter air masses were directly from urban areas of Guangzhou and
Foshan, indicating a strong influence from anthropogenic emissions. The temporal





variations of the total detected particles and oxalic acid containing particles in
summer and winter are shown in Figure 1. The total particles had similar trends with
the mass concentration of ambient $PM_{2.5}$, suggesting that the counts of total particles
detected by SPAMS can be representative of $PM_{2.5}$ mass concentration during the
whole sampling periods. The oxalic acid ($C_2$-containing) particles, in general,
exhibited distinct diurnal peaks from July 28 to August 1, while they showed different
temporal trends in winter. The relative abundance of oxalic acid particles in all of the
sampled particles ($C_2$/total ratio) had the same variations with the abundance of oxalic
acid particles in summer, especially in the period of July 28 – August 1 (Figure 1). In
winter, however, particle counts and relative abundance of oxalic acid had different
temporal changes except Jan 30 and February 5-8, when the count and relative
abundance of oxalic acid particles simultaneously had a sudden increase.
The oxalic acid-containing particles were clustered into eight groups, and they
altogether accounted for 89.6% and 95.1% of total oxalic acid particles in summer and
winter, respectively. Table 1 shows that in summer heavy metal (HM) type particles
contributed 31.3% to total oxalic acid particles, followed by the Sec (19.2%) and BB
type (13%). However, in winter BB type particles were the most abundant and
accounted for 24.2% of the oxalic acid-containing particles, followed by EC and HM
type. Besides, carbonaceous type particles including EC, OC, ECOC and BB
accounted for 28.1% of oxalic acid particles in summer and 59.8% in winter,
indicating the seasonal different characteristics of oxalic acid particles. The temporal
variations of eight groups of oxalic acid particles in summer and winter are illustrated
in Figure 1. In summer HM type particles (purple) and total oxalic acid particles
exhibited similar diurnal patterns, suggesting a possibly connection between the
production of oxalic acid and the transition metals (e.g. Fe, Cu) (Sorooshian et al.,
2013). Although Sec, BB and EC type particles showed similar diurnal patterns with
total oxalic acid particles, the concentrations of these type particles were generally
lower than HM type particles. In winter diurnal variation of oxalic acid particles was
not obvious but a sharp increase, accompanied by the increase of BB, EC and Sec
type particles, was observed on February 8.





The averaged positive and negative ion mass spectra of oxalic acid containing
particles are shown in Figure 2. The positive ion spectrum of oxalic acid particles in
summer was characterized by high fractions of metal ion peaks including $23[Na]^+$,
$27[Al]^+$, $39[K]^+$, $55[Mn]^+$, $56[Fe]^+$, $63/65[Cu]^+$,$64[Zn]^+$ and $208[Pb]^+$, and
carbonaceous marker ions at m/z $27[C_2H_3]^+$, $36[C_3]^+$, $43[C_2H_3O/C3H7]^+$, $48[C_4]^+$
(Figure 2 a). The negative ion spectrum of oxalic acid particles in summer was
characterized by the strong intensity of secondary ions including m/z $-46[NO_2]^-$,
$-62[NO_3]^-$, $-79[PO_3]^-$, $-80[SO_3]^-$, $-96[SO_4]^-$ and $-97[HSO_4]^-$, as well as carbon clusters
of $-24[C_2]^-$, $-36[C_3]^-$, $-48[C_4]^-$ and BB markers of $-59[C_2H_3O_2]^-$ and $-73[C_3H_5O_2]^-$
(Figure 2 b) (Zauscher et al., 2013). More carbonaceous clusters, i.e., $27[C_2H_3]^+$,
$29[C_2H_5]^+$, $36[C_3]^+$, $37[C_3H]^+$, $43[C_2H_3O]^+$, $48[C_4]^+$, $51[C_4H_3]^+$, $55[C_4H_7]^+$, $60[C_5]^+$,
$63[C_5H_3]^+$, $65[C_5H_5]^+$, $74[C_2H_2O_3]^+$, $77[C_6H_5]^+$, were observed in the positive ion
spectrum of oxalic acid particles in winter (Figure 2 c) than in summer. The negative
ion spectrum of oxalic acid particles in winter (Figure 2 d) contained a large amount
of secondary ions, similar to those found in summer, and a more intense signal of
nitric acid ($-125[HNO_3NO_3]^-$) , suggesting an acidic nature of oxalic acid particles in
winter.
The mixing ratios of oxalic acid particles with sulfate, nitrate and ammonium
(SNA) were investigated through the relative abundance of SNA-containing oxalic
acid particles in total oxalic acid particles (Figure 3). Oxalic acid was found to be
internally mixed with sulfate and nitrate during both sampling periods with mixing
ratio of 93% and 94% in summer respectively, and both 98% in winter (Figure 3 a).
However, the mixing ratio of $NH_4^+$ with oxalic acid was only 18% in summer but
increased to 71% in winter. Linear correlations between $NH_4^+$-containing oxalic acid
particles ($C_2$-$NH_4^+$) and total oxalic acid particles are depicted in Figure 3, with better
linear regression ($r^2$=0.98) in winter than summer. The low mixing ratio of $NH_4^+$ in
oxalic acid particles in summer indicated that the presence of oxalic acid in
$NH_4^+$-poor particles. Aqueous phase production of $SO_4^{2-}$ has been studied well and the
linear correlation between oxalic acid and $SO_4^{2-}$ has been used to study the production
of oxalic acid through aqueous phase reactions (Yu et al., 2005;Miyazaki et al.,





2009;Cheng et al., 2015). In our work, oxalic acid and $C_2$-$SO_4^{2-}$ displayed good
correlations in summer and winter (both $r^2$=0.99), which suggests a common
production route of oxalic acid and sulfate, likely aqueous phase reactions.

Figure 4 shows the unscaled size-resolved number distributions of the eight types

of oxalic acid particles. Oxalic acid mainly existed in 0.4 to 1.2 μm particles during
the entire sampling period but exhibited different peak modes for each particle type in
summer and winter. In summer, major types of oxalic acid particles showed distinct
peak mode at different size diameter. EC and Sec type particles peaked at 0.5 μm,
followed by BB type particles at 0.55 μm, then HM type particles at 0.6 μm, and OC
type particles at 0.7 μm. The difference of peak mode suggests possibly different
chemical evolution process for each type oxalic acid-containing particles. However, in
winter, oxalic acid particles showed broader size distribution from 0.5 to 0.8 μm for
all particle types. Oxalic acid particles of all types were generally larger in winter than
summer, possibly due to condensation and coagulation of particles during aging of
oxalic acid particles in winter.
**3.2 Photochemical production of oxalic acid in summer**

In summer oxalic acid particles showed peaks in the afternoon especially from

July 28 to August 1, which was in agreement with the variation pattern of the $O_3$
concentration (Figure 5), indicating a strong association of oxalic acid formation with
photochemical reactions. Malonic acid is another product of photochemical oxidation
of organic compounds (Kawamura and Ikushima, 1993;Wang et al., 2012;Meng et al.,
2013;Meng et al., 2014). In our campaign, malonic acid containing particles had
diurnal trends similar to oxalic acid particles and $O_3$ concentration. As the dominant
particle type, HM particles had identical variation pattern with total oxalic acid
particles. They are characterized by highly abundant metal ion peaks like 55$[Mn]^+$,
56$[Fe]^+$, 63/65$[Cu]^+$, 64$[Zn]^+$ and 208$[Pb]^+$, as well as secondary ion peaks of
-46$[NO_2]^-$, -62$[NO_3]^-$, -80$[SO_3]^-$, -96$[SO_4]^-$ and -97$[HSO_4]^-$ in the negative spectrum
in summer (Figure 6). •OH produced from Fenton reactions between $H_2O_2$ and $Fe^{2+/3+}$
in acidic solutions has been considered as a substantial source of •OH(Fenton,
1894;Dunford, 2002;Herrmann et al., 2015). The high abundance of metal ions in



oxalic acid particles may be an indication of possible Fenton reactions in the acidic
aqueous phase of acidic particles (pH<5, Figure S3), although we cannot exclude the
possibility of gas phase condensation of oxalic and malonic acids onto HM particles.
The oxidation of glyoxal and glyoxylic acid by •OH has been identified as an
important pathway of oxalic acid production by field and laboratory studies (Ervens et
al., 2004;Ervens and Volkamer, 2010;Wang et al., 2012). The modeling studies from
Ervens et al. (2014) suggest that oxalic acid production from glyoxal and glyoxylic
acid in aqueous phase significantly depends on •OH availability (Ervens et al., 2014).
While the partition of •OH from gas to aqueous phase is limited by its low Henry's
law constant ($K_{H,OH}$=30 M atm$^{-1}$) and short lifetime of •OH in the gas phase (Hanson
et al., 1992), the main sources of aqueous •OH are from the photolysis of $H_2O_2$, $NO_3^-$,
$NO_2^-$, and chromophoric dissolved organic matter (CDOM) (Yu et al., 2014;Badali et
al., 2015;Gligorovski et al., 2015;Tong et al., 2016). Among these sources the
photolysis of $H_2O_2$ through Fenton reactions involving the catalysis of transition
metal ions like $Fe^{2+/3+}$, $Cu^{+/2+}$ and $Mn^{2+/3+}$ is an efficient source of •OH (Deguillaume
et al., 2005;Herrmann et al., 2005;Ervens et al., 2014). The •OH formation process
through Fenton reactions can be expressed as (Ervens, 2015):
$Fe^{2+} + H_2O_2 \rightarrow FeOH^{2+} + \bullet OH$  (or $Fe^{3+} + OH^- + \bullet OH$)        (R1)
$FeOH^{2+}/Fe^{3+} + HO_2^{\cdot}/O_2^{\cdot-} \rightarrow Fe^{2+} + O_2 + H_2O/OH^-$                (R2)
The actual chemical process is far more complex and involves iron oxides and
iron-complexes, thus in the current work we focus on the potential availability of •OH
from Fenton reactions and the impact on the oxidation process of organic precursors.
In order to investigate the photochemical aqueous phase formation of oxalic acid
in summer, the diurnal variations of $O_3$, oxalic acid particles, HM group particles and
$pH_{is}$ of ambient particles averaged from July 28 to August 1, 2014 are shown in
Figure 7. The concentration of $O_3$ increased after 9:00 and peaked at 17:00, while
oxalic acid particles and HM group particles both increased after 10:00 and showed
two peaks at 15:00 and 19:00. The prominent photochemical feature of oxalic acid
particles suggested a close association of photochemical reactions with oxalic acid
production. Although •OH production from Fenton reactions can both occur under



dark and light radiation conditions, only photo-Fenton reactions had significant
contribution to the enhancement of oxalic acid particles in the current work. This was
possibly due to the diurnal variation of $pH_{is}$, since Fenton reactions strongly depend
on the pH of the aqueous phase (Gligorovski et al., 2015). When pH<1, $Fe^{2+}$ is
directly oxidized by $H_2O_2$ to $Fe^{3+}$ with no production of •OH (Barb et al.,
1951;Kremer, 2003), and the most favorable pH value for Fenton reaction is between
2.5 and 5 (Deguillaume et al., 2005). In the current work the $pH_{is}$ of ambient particles
ranged from -1.42 to 4.01, and the influences of $pH_{is}$ from RH and inorganic ions are
discussed in Figure S5. Strongly acidic particles were observed during the whole day
with high $pH_{is}$ at 6:00 and after 12:00. Although $pH_{is}$ was around 2 at 6:00, only a few
oxalic acid-containing particles were observed during this period due to low
abundance of HM particles. Oxalic acid-containing particles were found to increase
from 12:00 to 21:00, which was attributed to increased organic precursors from VOCs
oxidation and enhanced •OH production from Fenton reactions under $pH_{is}$ at 1-4. The
number concentration of oxalic acid particles peaked at 19:00 instead of during the
strong photochemical activity period in the afternoon; this was possibly due to the
efficient degradation of oxalic acid from the complex of $Fe(oxalate)_2^-$ (Sorooshian et
al., 2013;Zhou et al., 2015). On the other hand, photolysis of $Fe(oxalate)_2^-$ can
contribute to 99% of the overall degradation of oxalic acid (Weller et al., 2014).
Although the enhanced •OH production from photo-Fenton reactions was favorable
for the formation of oxalic acid from 12:00 to 18:00, we speculate that a high
degradation rate of oxalic acid by iron complexation resulted in a lower net
production of oxalic acid than at 19:00.

Based on above discussions, detailed mechanism for oxalic acid formation in

acidic aqueous phase of particles is proposed for our field observations (Figure 8). In
summer strong photochemical activity and high $O_3$ concentrations in the afternoon
leads to more production of reactive radicals such as •OH and $HO_2^•$, which promote
the oxidation of VOCs to dicarbonyls and aldehydes (e.g. glyoxal and methylglyoxal),
followed by a subsequent partitioning into the aqueous phase of particles
(Myriokefalitakis et al., 2011). Acidic particles containing transition metals like Fe



and Cu potentially yield more •OH in acidic aqueous phase, then hydrated dicarbonyls
and aldehydes can be oxidized by •OH to glyoxylic acid and finally to oxalic acid
(Wang et al., 2012). Recently Ma et al. (2015) had studied the Fe-containing particles
in the PRD and found Fe-containing particles are more efficient at generating •OH in
summer than winter (Ma et al., 2015), which supports the enhanced •OH production
in HM type particles in this work. A large amount of Fe related particles are emitted
from steel industries in the North China Plain and metals like V, Zn, Cu and Pb from
electronic manufacturing (Cui and Zhang, 2008;Dall'Osto et al., 2008). These metals
contribute significantly to haze episodes (Moffet et al., 2008;Li et al., 2014), which
possibly increases the formation of SOA by yielding more OH participating the
heterogeneous and aqueous reactions.

### 3.3 Formation process of oxalic acid in winter

Despite lower $O_3$ concentrations and photochemical activity in winter, oxalic
acid particles were still prevalent in carbonaceous particles, especially BB type
particles. The sharp increase of oxalic acid particles on February 8, 2015 (Figure 1)
was selected as a typical episode to investigate the formation processes of oxalic acid
in winter.
During the episode, the 48 hr back trajectory analysis showed air masses that
originated from the urban areas of Guangzhou and Foshan city (Figure S4), indicating
strong influence on organic precursors from anthropogenic emissions. Oxalic acid
particle types were dominated by BB (23.2%), followed by EC (22.0%) and Sec
(15.1%) type (Table 2). Carbonaceous particles including EC, ECOC, OC, BB
accounted for 61.6% of the total oxalic acid particles. The mass spectra of oxalic acid
particles were characterized by many hydrocarbon clusters of $27[C_2H_3]^+$, $29[C_2H_5]^+$,
$37[C_3H]^+$, $43[C_2H_3O]^+$, $51[C_4H_3]^+$, $55[C_4H_7]^+$, $63[C_5H_3]^+$, $65[C_5H_5]^+$, $74[C_2H_2O_3]^+$,
$77[C_6H_5]^+$, and carbon clusters of $36[C_3]^+$, $48[C_4]^+$, $60[C_5]^+$ in positive mass spectrum,
while the negative mass spectrum was characterized by elemental carbon clusters like
$-24[C_2]^-$, $-36[C_3]^-$, $-48[C_4]^-$, biomass burning markers of $-59[C_2H_3O_2]^-$ and
$-73[C_3H_5O_2]^-$ and secondary species including $-42[CNO]^-$, $-46[NO_2]^-$, $-62[NO_3]^-$,
$-79[PO_3]^-$, $-80[SO_3]^-$, $-96[SO_4]^-$ and $-97[HSO_4]^-$ (Figure 9 a). The nitric acid was



identified in oxalic acid particles not only in the episode but also during the entire
sampling period in winter, indicating a strongly acidic nature of oxalic acid particles
in winter.
As the precursor of oxalic acid, glyoxal has the potential to react with sulfuric
acid to produce organosulfates through acid-catalyzed nucleophilic addition according
to laboratory and chamber studies(Surratt et al., 2007;Galloway et al., 2009). The
negative ion of -155($[C_2H_3O_2SO_4]^-$) has been identified as the marker ion of
organosulfates derived from glyoxal in chamber and field measurements using
ATOFMS (Surratt et al., 2008;Hatch et al., 2011). The organosulfate derived from
glyoxal requires acidic aqueous environment of particles, and herein is used as an
indicator of acid-catalyzed ageing process of organic compounds. The temporal
variation of organosulfate (m/z=-155) containing particles during the entire sampling
period in Heshan, China is shown in Figure S6. During the episode, oxalic acid
particles had moderate linear correlation with organosulfate particles (Figure 9b).
Based on the above discussion, the degradation of carbonaceous species associated
with acid-catalyzed reactions may have a significant contribution to the formation of
oxalic acid during the episode in winter. Similar particle types and mass spectra of
oxalic acid-containing particles during the episode and the whole sampling period in
winter were observed, which suggest the acid-catalyzed oxidation of organic
precursors as a potential source for oxalic acid.

## 4. Summary and conclusions

Oxalic acid containing particles were measured by a single particle aerosol mass
spectrometer (SPAMS) in the summer and winter of 2014 in Heshan, China. They
accounted for 2.5% and 2.7% of the total detected ambient particles. In summer heavy
metal-containing particles were the largest group of particles containing oxalic acid
with a fraction of 31.3% followed by Sec type (19.2%), while in winter BB type was
the dominant group with a percentage of 24.2%. More than 90% of oxalic acid
particles were internally mixed with sulfate and nitrate during the whole sampling
period. Only 18% of oxalic acid particles contained ammonium in summer, which





increased to 71% in winter. In summer oxalic acid and $O_3$ concentration exhibited
similar diurnal variations, indicating a substantial contribution of photochemical
reactions to oxalic acid formation. The favorable in-situ pH and the dominance of
transition metal ions in oxalic acid particles suggests an enhanced production of •OH
from Fenton like reactions. A mechanism involving the photochemical production of
VOCs via efficient aqueous phase reactions with enhanced •OH to oxalic acid was
proposed. In winter carbonaceous type particles including EC, OC, ECOC and BB
groups accounted for 59.8% of oxalic acid particles and increased to 61.6% in the
episode. Nitric acid and organosulfate were found to co-exist in oxalic acid-containing
particles in the winter, which suggests a close association with acid-catalyzed
reactions. Acid-catalyzed oxidation of organic precursors is a potential contribution
for the formation of oxalic acid in winter. The current study also indicates that
SPAMS can be a robust tool for exploring the formation and transformation processes
of SOA, contributing to the improvement of global climate modeling and the
development of effective air pollution mitigation strategies.

## Acknowledgments

This work was financially supported by National Key Technology R&D Program
(Grant No. 2014BAC21B01), Guangdong Province Public Interest Research and
Capacity Building Special Fund (Grant No. 2014B020216005), the Strategic Priority
Research Program (B) of the Chinese Academy of Sciences (Grant No.
XDB05040502), Guangdong Industry-University Research Program (Grant
No.2012B090500014), and NSFC of Guangdong Province (Grant No.
2015A030313339). Chak K. Chan would like to acknowledge funding support of the
General Fund of National Natural Science Foundation of China (Grant No.

41675117).

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





**Tables and Figures**

**Table list:**

Table 1. Summary of major groups of oxalic acid-containing particles in summer and winter in PRD, China.

Table 2. The abundance of major particle types in total oxalic acid-containing particles during the episode in winter (2/8/2015).

**Figure caption:**

Figure 1. Temporal variations of total detected particles and oxalic acid containing particles during whole sampling periods in Heshan, China: (a) hourly variations of $PM_{2.5}$ mass concentration, total detected particle counts, oxalic acid containing particles, ratio of oxalic acid-containing/total particles and major types of oxalic acid containing particles; (b) variation patterns of relative abundance of major types of oxalic acid containing particles.

Figure 2. The averaged positive and negative ion mass spectra of oxalic acid containing particles is investigated in summer and winter: (a) summer positive, (b) summer negative, (c) winter positive, (d) winter negative. The color bars represent each peak area corresponding to specific fraction in individual particles.

Figure 3. (a) Mixing state of oxalic acid with sulfate, nitrate and ammonium in oxalic acid-containing particles; (b) Linear correlation between $NH_4^+$-containing oxalic acid particles and the total oxalic acid particles in summer; (c) Linear correlation between $NH_4^+$-containing oxalic acid particles and the total oxalic acid particles in winter. Abbreviations: $C_2$-$NH_4^+$ represents the $NH_4^+$-containing oxalic acid particles, and same expressions for $C_2$-$SO_4^{2-}$ and $C_2$-$NO_3^-$.





Figure 4. Unscaled size-resolved number distributions of major types of oxalic acid
particles in summer and winter.

Figure 5. Temporal variations of $O_3$ concentrations, oxalic acid particles, malonic acid
particles and heavy metal type of oxalic acid particles during the entire sampling
period in Heshan, China.

Figure 6. The averaged digitized positive and negative ion mass spectra of heavy
metal type of oxalic acid-containing particles in summer.

Figure 7. The diurnal variations of $O_3$ concentration, oxalic acid particles, HM group
particles and in-situ pH ($pH_{is}$) from July 28 to August 1 in 2014.

Figure 8. The formation process of oxalic acid in the aqueous phase of particles in
summer: the red steps are enhanced by photochemical activities in the current study.

Figure 9. The comprehensive study of oxalic acid particles increase on Feb 8, 2015: (a)
The digitized positive and negative ion mass spectrum of oxalic acid particles during
the episode; (b) Linear regression between oxalic acid particles and organosulfate
particles (m/z -155).

















Table 1. Summary of major groups of oxalic acid-containing particles in summer and winter in PRD, China.

| Particle type | Summer(7/18-8/1, 2014) | | Winter(1/27-2/8, 2015) | |
|---|---|---|---|---|
| | Count | Percentage, % | Count | Percentage, % |
| EC | 1473 | 11.2 | 3161 | 15.4 |
| ECOC | 41 | 0.3 | 2233 | 10.9 |
| OC | 473 | 3.6 | 1922 | 9.4 |
| BB | 1702 | 13.0 | 4953 | 24.2 |
| HM | 4104 | 31.3 | 3124 | 15.2 |
| Sec | 2511 | 19.2 | 2192 | 10.7 |
| NaK | 303 | 2.3 | 17 | 0.1 |
| Dust | 1139 | 8.7 | 1888 | 9.2 |

Abbreviations of major particle types: elemental carbon (EC), elemental and organic carbon (ECOC), organic carbon (OC), biomass burning (BB), heavy metal (HM), secondary (Sec), sodium and potassium (NaK) and dust (Dust).






Table 2. The abundance of major particle types in total oxalic acid-containing particles during the episode in winter (2/8/2015).

| | EC | ECOC | OC | BB | Sec | HM | Dust | other |
|---|---|---|---|---|---|---|---|---|
| Count | 1250 | 604 | 326 | 1320 | 856 | 377 | 814 | 132 |
| Percentage, % | 22.0 | 10.6 | 5.7 | 23.2 | 15.1 | 6.6 | 14.3 | 2.3 |


















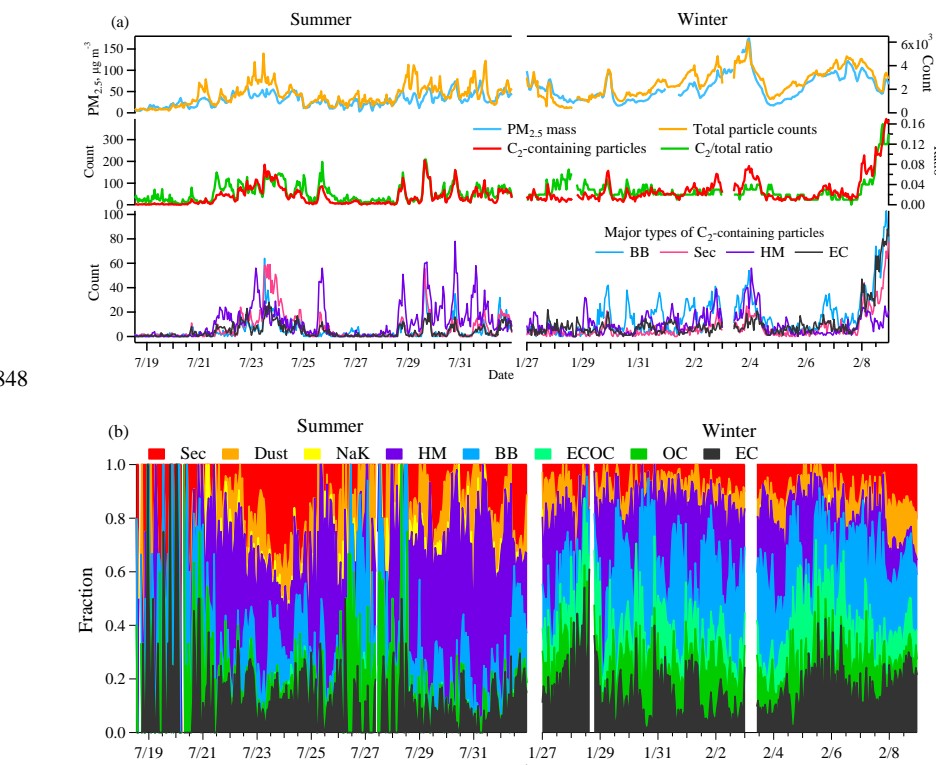



Figure 1. Temporal variations of total detected particles and oxalic acid containing
particles during whole sampling periods in Heshan, China: (a) hourly variations of
PM$_{2.5}$ mass concentration, total detected particle counts, oxalic acid containing
particles, ratio of oxalic acid-containing/total particles and major types of oxalic acid
containing particles; (b) variation patterns of relative abundance of major types of
oxalic acid containing particles. Abbreviations of major particle types: elemental
carbon (EC), organic carbon (OC), elemental and organic carbon (ECOC), biomass
burning (BB), heavy metal (HM), secondary (Sec), sodium and potassium (NaK) and
dust.










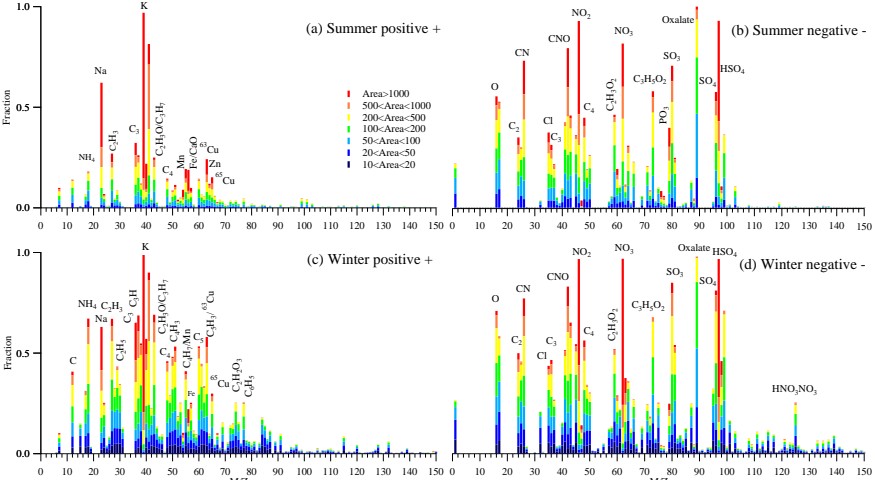


Figure 2. The averaged positive and negative ion mass spectra of oxalic acid containing particles is investigated in summer and winter: (a) summer positive, (b) summer negative, (c) winter positive, (d) winter negative. The color bars represent each peak area corresponding to specific fraction in individual particles.




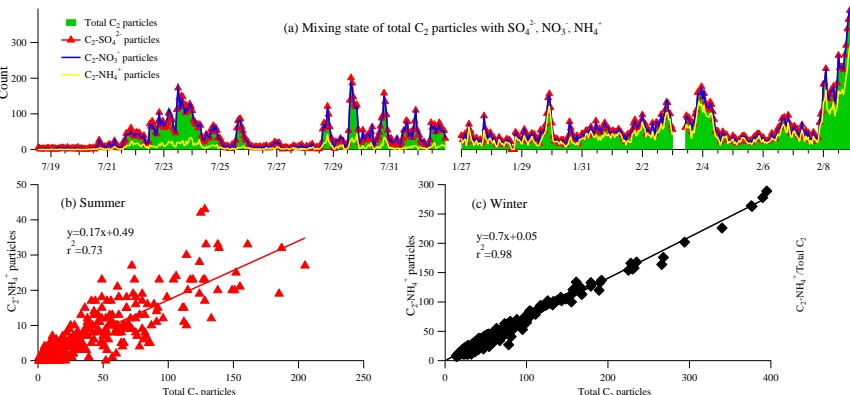


Figure 3. (a) Mixing state of oxalic acid with sulfate, nitrate and ammonium in oxalic
acid-containing particles; (b) Linear correlation between $NH_4^+$-containing oxalic acid
particles and the total oxalic acid particles in summer; (c) Linear correlation between
$NH_4^+$-containing oxalic acid particles and the total oxalic acid particles in winter.
Abbreviations: $C_2$-$NH_4^+$ represents the $NH_4^+$-containing oxalic acid particles, and
same expressions for $C_2$-$SO_4^{2-}$ and $C_2$-$NO_3^-$.


















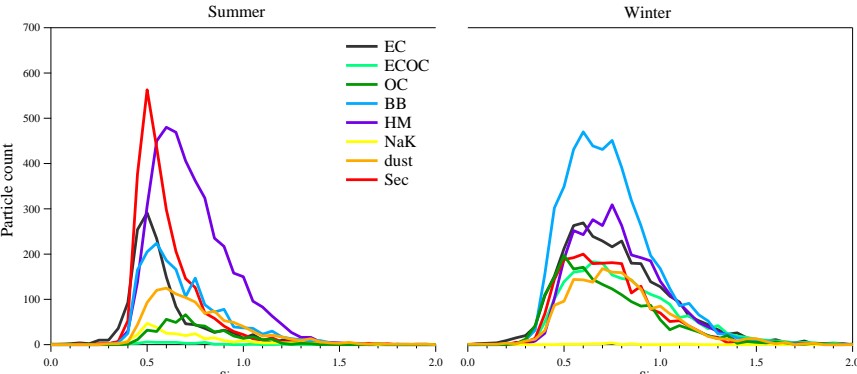


Figure 4. Unscaled size-resolved number distributions of major types of oxalic acid
particles in summer and winter. Abbreviations of major particle types: elemental
carbon (EC), organic carbon (OC), elemental and organic carbon (ECOC), biomass
burning (BB), heavy metal (HM), secondary (Sec), sodium and potassium (NaK) and
dust.

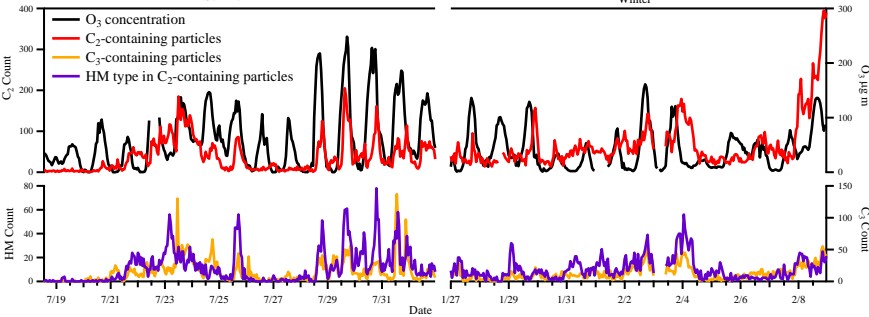


Figure 5. Temporal variations of $O_3$ concentrations, oxalic acid particles, malonic acid
particles and heavy metal type of oxalic acid particles during the entire sampling
period in Heshan, China.

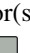


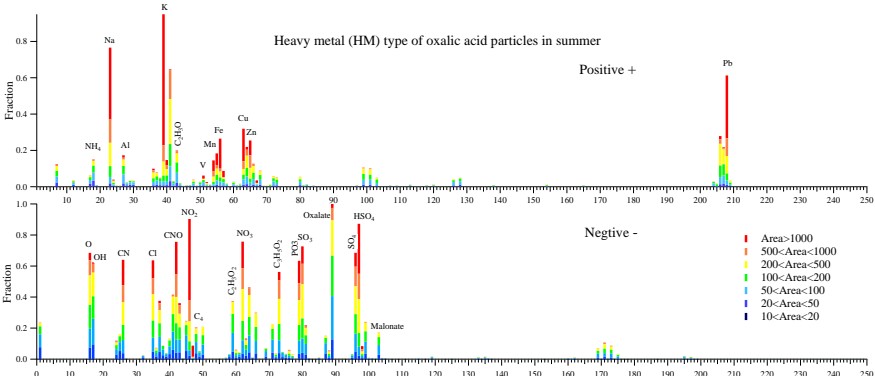


Figure 6. The averaged digitized positive and negative ion mass spectra of heavy

metal type of oxalic acid-containing particles in summer.


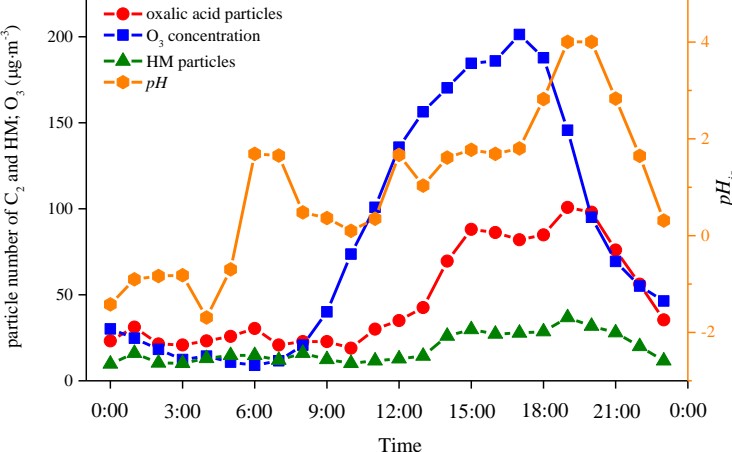


Figure 7. The diurnal variations of $O_3$ concentration, oxalic acid particles, HM group

particles and in-situ pH ($pH_{is}$) from July 28 to August 1 in 2014.







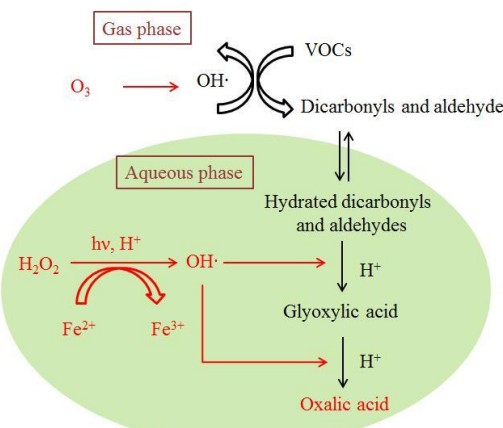


Figure 8. The formation process of oxalic acid in the aqueous phase of particles in
summer: the red steps are enhanced by photochemical activities in the current study.






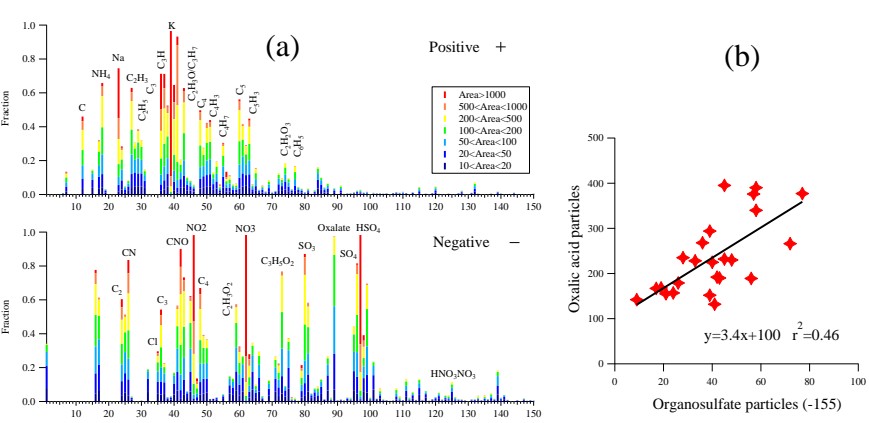


Figure 9. The comprehensive study of oxalic acid particles increase on Feb 8, 2015: (a)
The digitized positive and negative ion mass spectrum of oxalic acid particles during
the episode; (b) Linear regression between oxalic acid particles and organosulfate
particles (m/z -155).
