# Peer review of "Mixing state of oxalic acid containing particles in the rural area of Pearl"

_Atmospheric Chemistry and Physics, 2016_

## Referee Comment (RC1) · Anonymous Referee #1 · 9 Jan 2017

Review for "Mixing state of oxalic acid containing particles in the rural area of Pearl River Delta, China: implication for seasonal formation mechanism of secondary organic aerosol (SOA" by Chunlei Cheng et al.

General comments: This paper reports the observation results of abundances and mixing states of oxalic acid containing particles using a single particle aerosol mass spectrometer in the summer and winter of 2014 in Heshan, an atmospheric measurement supersite of PRD region, China. Based on the difference in mass spectrometry, the authors proposed two different seasonal formation mechanisms for oxalic acid. This

work presented interesting results, which is helpful for improving our understanding on SOA formation. After addressing the questions below, this paper could be accepted for a final publication in the journal.

Detailed comments: 1. Page 3, line 84-91, these statements are confusing. On one hand, the authors claimed that primary emissions of DCA are minor. On the other hand, however, they stated that high concentrations of DCA have been observed in biomass burning plumes and the primary sources are still unclear. So, the authors should clarify if the primary source of DCA is important or not. 2. Page 7, line 189-193, what principle here is used for the particle type definitions? It is hard to understand what are the differences between the ECOC type, EC and OC. In Table S1, the authors presented the criteria for particle type identification, is there any references to support such definitions? 3. Page 7, line 201-209, as for the ISORROPIA model, which mode you used? stable mode or metastable mode? What is the basis of the mode you choose? Since this work claimed that there are at least eight types of oxalic acid-containing particles, which means that aerosols measured here can not be taken as a single phase. In other words, the ISORROPIA mode should be run as a stable mode. Did the authors run this mode for the acidity and ALWC calculation? Please clarify. 4. Page 7, line 211-216, please give the locations of Guangzhou and Foshan in Figure S4. 5. Page 8, line 239, in Figure 1a, the color of HM type particles is blue not purple? 6. Page 10, section 3.2. Zn and Pb are typical tracers for vehicle emissions. The strong associations between heavy metals and oxalic acid in summer can be ascribed to the dominant emissions from vehicle exhausts. Oxalic acid can be produced via lots of formation pathways. Do you have direct evidence on the Fenton-like reaction? In the abstract section, authors stated that during summer ozone and oxalic acid-containing particles presented a similar temporal variation pattern, which indicates that photo-chemical reaction is important for oxalic acid production. Thus, as for the above two formation mechanisms, which is more important for oxalic acid?

---

## Referee Comment (RC2) · Anonymous Referee #2 · 11 Jan 2017

The authors present a comprehensive analysis of the mixing state of oxalic acid in the Pearl River Delta area of China using a single particle aerosol mass spectrometer (SPAMS). The topic of the paper is important with regard to understanding formation pathways of secondary organic aerosol (SOA). The main findings are the following: oxalic acid containing particles accounted for <3% of total particles; in summer heavy-metals containing particles were the largest group containing oxalic acid while in winter it was the biomass burning group; the majority of oxalic acid particles were internally mixed with sulfate and nitrate; the fraction of oxalic acid particles containing ammonium increases significantly in winter versus summer. A couple of interesting speculations

are made about acid-catalyzed reactions and Fenton like reactions.

The paper is written well. The title and abstract are mostly reflective of the contents of the paper. The results are quite interesting and I support publication after each of my comments below are adequately addressed:

Specific Comments: While the title of the paper and parts of the manuscript make reference to comparison of summer and winter and also mention the word 'seasonal', it is important to consider that the authors are only looking at two short-term periods spanning $\sim$2 weeks. I suggest a relaxation of words in parts of the paper that make it seem as though full seasonal coverage was obtained. At the minimum, 'seasonal' needs to be revised in my view for the title.

Line 189: These percentages are quite interesting to me and they seem low based on how ubiquitous the literature suggests oxalic acid is in particles. Can the authors comment more about how these percentages compare to previous reports?

Line 189-193: While the authors refer readers to the Supplement, it would be useful here to at least provide a little more detail as to how the categorization was done assuming that not all readers will go to the Supplement. The categories are very important for the results, so some more discussion is warranted here as to how this was done.

Line 289-291: Could the lower mixing height in the winter have contributed in some way to this finding (i.e. more stagnant aerosol and perhaps more aged)?

Line 337-338: The wording here in this sentence and the general paragraph appear to be too strong in my view since the authors did not unambiguously prove that photo-Fenton reactions are even occurring. Aren't these just speculations? I suggest to use less strong language and to differentiate better between proved findings and speculations.

Line 329-359: have the authors considered all other factors that could affect the diurnal

behavior of oxalic acid particles, such as meteorological factors or different emissions types during the day. Can the authors comment on how the profiles of other species look in Figure 7 if they were plotted such as sulfate, EC, nitrate? Is rush hour traffic influential at all in any of the discrepancies between the peak of ozone and the other parameters currently shown in Figure 7? While these other factors may not be important, it is still important to mention that various other potential factors were considered.

Section 3.3: the discussion and analysis surrounding the acid-catalyzed hypothesis is too thin in my view. Were such relationships not observed in the summer, and if not, then why? Could another reasonable explanation be that that precursors of oxalic acid and the organosulfate species are co-emitted? To strengthen this conclusion, the discussion and analysis needs to be more convincing with also more discussion of other relevant words using field data to point to this mechanism of acid-catalyzed formation of SOA.

Line 107-110: This line needs a revision because it is not entirely fair. It should be noted somewhere around this section that quite a bit of work has been done with fast time resolution on aircraft to address the issue of meteorological uncertainty and temporal resolution limitations. These various studies have discussed the formation pathways leading to oxalic acid with detailed in-cloud and out-of-cloud measurements:

Wonaschuetz, A., et al. (2012). Aerosol and gas re-distribution by shallow cumulus clouds: an investigation using airborne measurements, J. Geophys. Res., 117, D17202, doi:10.1029/2012JD018089.

Sorooshian, A., et al. (2006). Oxalic acid in clear and cloudy atmospheres: Analysis of data from International Consortium for Atmospheric Research on Transport and Transformation 2004, J. Geophys. Res. 111, D23S45, doi:10.1029/2005JD006880.

Sorooshian, A., et al. (2007). Particulate organic acids and overall water-soluble aerosol composition measurements from the 2006 Gulf of Mexico Atmospheric Composition and Climate Study (GoMACCS), J. Geophys. Res., 112, D13201,

doi:10.1029/2007JD008537.

Figures: Font size needs to increase in many of the figures for labels.

———————————————————

---

## Referee Comment (RC3) · Anonymous Referee #3 · 19 Jan 2017

The authors present results measurements of ambient aerosol during two periods (summer vs winter) in the Pearl River Delta. Most of the discussion focusses on oxalate loadings and its mixing state. Based on correlations with metal-containing particles, the authors conclude that iron has an important role in OH formation and therefore in oxalate formation. While generally the manuscript is well written, the discussion of the chemical mechanism is very weak and hand waving. Without running a detailed multiphase model, such conclusions cannot be drawn with certainty. Therefore, I suggest removing the discussion of the role of iron and the chemical mechanism in general. After my additional comments below are addressed, the manuscript may be suitable

for publication.

Major comments

1) The suggested chemical mechanism is oversimplified and contains several misconceptions and/or omissions:

a) Generally, it is assumed that direct OH uptake is the main source of OH in the aqueous phase [e.g., Ervens et al., 2003; Herrmann, 2003; Tilgner et al., 2013]. Therefore higher OH(gas) concentration will lead to higher OH concentration in the aqueous phase. Higher OH(aq) concentration will also lead to more oxidation of oxalate and therefore less SOA.

b) Higher iron concentration might lead to more OH. However, more importantly is the effect of the loss of oxalate due to the photolysis of the iron-oxalato complex. While this reaction is mentioned in the manuscript, its predominating role in oxalate loss [Sorooshian et al., 2013] is not discussed in a balanced way.

c) At low pH, it can be expected that reaction rates are lower since in general the undissociated acids (glyoxylic, glycolic) react more slowly than their dissociated counterparts. Oxalate has a very low pKa (1.23) so that even at low pH a substantial fraction is still present as oxalate. Could changes in pH and therefore reaction rates explain some of the temporal trends?

d) At very low pH (< 1.23), it is expected that oxalic acid is present in undissociated form and therefore not able to make salts or complexes that 'trap' it in the particle phase. This fact contradicts the trend of increased oxalate concentrations at low pH. This should be discussed.

e) The proposed mechanism is by no means new or detailed (l. 360). It does not include any sinks of oxalate, nor complex formation. It is one possible formation mechanism of oxalic acid from glyoxal. The generalization to dicarbonyls and aldehydes is not correct since only small compounds (C2) will follow the suggested reaction pathways.

2) The number fraction of oxalate containing particles seems very low. Is this comparable to other measurements? What was the mass fraction of oxalate (a) in the particles and (b) related to the total aerosol loading?

3) Was all iron in the particles in form of soluble iron, i.e. available for reaction?

4) Oxalate and the other DCAs usually represent only a very small fraction of the total organic aerosol mass. Therefore the title is misleading as it talks about SOA in general.

5) It seems based on Figure S5, that RH was always < 100% (except a very brief period). Therefore, the discussed aqueous chemistry will have to take place in aqueous aerosol. There are many studies that have discussed different reaction pathways in aqueous aerosol vs cloud [e.g., Tan et al., 2009; Lim et al., 2010] with less efficient oxalate formation in the former. In addition, it seems likely that iron ions might be less dissolved in the rather highly concentrated aqueous aerosol solutions. All discussion is about chemistry as it happens in cloud droplets. These two regimes should be differentiated.

6) While briefly discussed, it is not clear to what extent different air masses cause different oxalate levels. How much of the measured oxalate is background material? Did other meteorological conditions affect the concentrations such as changes in boundary layer? 7) I am not sure what Figure S6 is really showing. Does it show a correlation of organosulfur particles and oxalate or does it simply show that more particles cause higher concentrations of 'everything'? How about the mixing state of organosulfur compounds and oxalic acid particles? The fact that they are in the same particle class, does not necessarily mean that they are internally mixed and therefore their formation pathways are related.

Minor comments

l. 83: Oxalic acid does not have a low vapor pressure. Its presence in ambient particles

is due to salt and/or complex formation (cf also comment 1d).

l. 119/120: There are several studies that have shown good agreement between predicted and measured oxalate levels [e.g., Wonaschuetz et al., 2012]

l. 121 and 122: These sentences are repetitive.

l. 240: The study by Sorooshian et al. focused mostly on the destruction of oxalalte in the presence of iron.

l. 264, and other places: 'Mixing ratio' usually refers to the ratio of molecules of one type to the total number of molecules (e.g. ppb = 1 in 10ˆ9 molecules). The authors should change their wording as I assume here 'mixing ratio' is used in the meaning of 'number of particles that are internally mixed'.

l. 289: Are all particle larger in winter than in summer or only those that contain oxalic acid?

l. 296-300: The mentioning of malonic acid is distracting here and does not lead to additional evidence or insights.

l. 316: It is true that OH (like all other radicals) has a relatively short life time in the gas phase. However, the partitioning to the aqueous phase is limited due to its even shorter lifetime in the aqueous phase. Its solubility and the quick consumption in the aqueous phase leads to the limitation.

l. 320: Fenton reaction is not a photolysis. In l. 337, it is stated correctly that Fenton reactions do not need necessarily light.

l. 344: What are the influences of pH(is) from RH and inorganic ions? Figure S5 does not include any discussion.

l. 354: Not clear why 'on the other hand' as the following sentence is just another example of oxalate degradation.

l. 439/440: This is a very strong and vague statement. How do the results of the current study help improving climate models and air pollution mitigation strategies?

Technical comments

l. 102: replace 'suggest' by 'suggested'

l. 181: replace 'introducing' by 'introduced'

l. 240: replace 'possibly' by 'possible'

l. 273: remove 'that'

References

Ervens, B., C. George, J. E. Williams, G. V. Buxton, G. A. Salmon, M. Bydder, F. Wilkinson, F. Dentener, P. Mirabel, R. Wolke, and H. Herrmann (2003), CAPRAM2.4 (MODAC mechanism): An extended and condensed tropospheric aqueous phase mechanism and its application, J. Geophys. Res., 108(D14), 4426, doi: 10.1029/2002JD002202.

Herrmann, H. (2003), Kinetics of aqueous phase reactions relevant for atmospheric chemistry, Chem. Rev., 103(12), 4691-4716.

Lim, Y. B., Y. Tan, M. J. Perri, S. P. Seitzinger, and B. J. Turpin (2010), Aqueous chemistry and its role in secondary organic aerosol (SOA) formation, Atmos. Chem. Phys., 10(21), 10521-10539.

Sorooshian, A., Z. Wang, M. M. Coggon, H. H. Jonsson, and B. Ervens (2013), Observations of Sharp Oxalate Reductions in Stratocumulus Clouds at Variable Altitudes: Organic Acid and Metal Measurements During the 2011 E-PEACE Campaign, Environ. Sci. Technol., 47(14), 7747-7756, 10.1021/es4012383.

Tan, Y., M. J. Perri, S. P. Seitzinger, and B. J. Turpin (2009), Effects of Precursor Concentration and Acidic Sulfate in Aqueous Glyoxal-OH Radical Oxidation and Implications for Secondary Organic Aerosol, Environ. Sci. Technol., 43(21), 8105-8112,

10.1021/es901742f.

Tilgner, A., P. Bräuer, R. Wolke, and H. Herrmann (2013), Modelling multiphase chemistry in deliquescent aerosols and clouds using CAPRAM3.0i, J. Atmos. Chem., 70(3), 221-256, 10.1007/s10874-013-9267-4.

Wonaschuetz, A., A. Sorooshian, B. Ervens, P. Y. Chuang, G. Feingold, S. M. Murphy, J. de Gouw, C. Warneke, and H. H. Jonsson (2012), Aerosol and gas re-distribution by shallow cumulus clouds: An investigation using airborne measurements, J. Geophys. Res. - Atmos., 117(D17), D17202, 10.1029/2012jd018089.

---

## Author Comment (AC1) · 6 Apr 2017

**Response to the comments of Anonymous Referee #1**

[Atmospheric Chemistry and Physics, MS ID: acp-2016-1081]
Title: Mixing state of oxalic acid containing particles in the rural area of Pearl River Delta, China: implication for seasonal formation mechanism of Secondary Organic Aerosol (SOA)

**General comments:**
This paper reports the observation results of abundances and mixing states of oxalic acid containing particles using a single particle aerosol mass spectrometer in the summer and winter of 2014 in Heshan, an atmospheric measurement supersite of PRD region, China. Based on the difference in mass spectrometry, the authors proposed two different seasonal formation mechanisms for oxalic acid. This work presented interesting results, which is helpful for improving our understanding on SOA formation. After addressing the questions below, this paper could be accepted for a final publication in the journal.

**Response**: Thank you for your comments. These comments are all valuable and very helpful for revising and improving our paper, as well as the important guiding significance to our researches. We have studied these comments carefully and have made corrections. Our responses to the comments are itemized below.

Anything about our paper, please feel free to contact me at limei2007@163.com

Best regards!

Sincerely yours

Mei Li
April 6, 2017

**Specific comments and point by point responses:**

1. Page 3, line 84-91, these statements are confusing. On one hand, the authors claimed that primary emissions of DCA are minor. On the other hand, however, they stated that high concentrations of DCA have been observed in biomass burning plumes and the primary sources are still unclear. So, the authors should clarify if the primary source of DCA is important or not.

**Response**: The primary emissions of DCAs from anthropogenic sources in urban areas are minor according to reported studies (Huang and Yu, 2007;Stone et al., 2010). As for the biomass burning emission, research work from biomass burning plume showed less than 23% of oxalic acid had strong connection with direct emission from biomass burning(Kundu et al., 2010), so the primary source of DCAs is less important compared to secondary sources. We have changed "Even though high concentrations of DCAs have been observed in air masses influenced by biomass burning (Kundu et al., 2010;Kawamura et al., 2013), the primary source of DCAs is still not clear (van Pinxteren et al., 2014)." to "High concentrations of DCAs have been observed in biomass burning plume (Kundu et al., 2010;Kawamura et al., 2013) with more than 70% of DCAs produced from photochemical oxidation of water-soluble organic compounds, and only a small contribution from direct biomass burning emission (van Pinxteren et al., 2014)." in lines 89-92.

2. Page 7, line 189-193, what principle here is used for the particle type definitions? It is hard to understand what are the differences between the ECOC type, EC and OC. In Table S1, the authors presented the criteria for particle type identification, is there any references to support such definitions?

**Response**: We have added the classification rules for oxalic acid particles in the manuscript and explain the differences among the EC, OC, ECOC type, and several literature publications are cited in the manuscript to support these classification rules. "All oxalic acid particles are classified into eight types: elemental carbon (EC), organic carbon (OC), elemental and organic carbon (ECOC), biomass burning (BB), heavy metal (HM), secondary (Sec), sodium-potassium (NaK) and dust." has been changed to "The oxalic acid containing particles are classified into eight types in the following order: elemental carbon (EC), organic carbon (OC), elemental and organic carbon (ECOC), biomass burning (BB), heavy metal (HM), secondary (Sec), sodium-potassium (NaK) and dust. Different type particles are identified according to characteristic ion markers and dominant chemical species (Table S1): (1) particles containing abundant carbon clusters like $\pm12[C]^{+/-}$, $\pm24[C_2]^{+/-}$, $\pm36[C_3]^{+/-}$ with relative peak area more than 0.5% are classified as EC type, (2) any remaining particles containing abundant signals of $27[C_2H_3]^+$, $43[C_2H_3O]^+$ and hydrocarbon clusters with relative peak area more than 0.5% are classified as OC type, (3) any remaining particles containing signals of $\pm12[C]^{+/-}$, $\pm24[C_2]^{+/-}$, $37[C_3H]^+$ and $43[C_2H_3O]^+$ with relative peak area more than 0.5% are classified as ECOC type, (4) any remaining particles containing abundant signals of $39[K]^+$ (peak area>1500) with relative peak area of $-59[C_2H_3O_2]^-$ and $-73[C_3H_5O_2]^-$ simultaneously more than 0.5% are classified

as BB type, (5) any remaining particles containing signals of 55[Mn]$^+$, 56[Fe]$^+$, 63/65[Cu]$^+$, 64[Zn]$^+$ and 208[Pb]$^+$ with relative peak area more than 0.5% are classified as HM type, (6) any remaining particles containing abundant signals of 18[NH$_4$]$^+$ (peak area>50), -62[NO$_3$]$^-$ (peak area>100) and -97[HSO$_4$]$^-$ (peak area>100) are classified as Sec type, (7) any remaining particles containing abundant signals of 23[Na]$^+$ (peak area>1500) and related species are classified as NaK type, (8) any remaining particles containing signals of 40[Ca]$^+$, 56[CaO]$^+$ and related species are classified as dust type. The rules for oxalic acid particles classification in the current work have been reported in previous studies (Sullivan and Prather, 2007; Yang et al., 2009; Zhang et al., 2013; Li et al., 2014)." in lines 199-221.

3. Page 7, line 201-209, as for the ISORROPIA model, which mode you used? stable mode or metastable mode? What is the basis of the mode you choose? Since this work claimed that there are at least eight types of oxalic acid-containing particles, which means that aerosols measured here can not be taken as a single phase. In other words, the ISORROPIA mode should be run as a stable mode. Did the authors run this mode for the acidity and ALWC calculation? Please clarify.

**Response**: We have used stable mode and reverse type in the ISORROPIA model to calculate the concentration of H$^+$ and ALWC. Then we calculated the in-situ pH based on the below equation:

$$pH_{is} = -\log\alpha_{H^+} = -\log(\gamma_{H^+} \times n_{H^+} \times 1000/V_a)$$ , where $n_{H^+}$ is the concentration of H$^+$

(mol m$^{-3}$) and $V_a$ is the volume concentration of the H$_2$O (cm$^3$ m$^{-3}$). In order to classify this point, we add the description of the running mode and type in the ISORROPIA mode. "We choose stable mode and reverse type in the ISORROPIA model to calculate the concentration of H$^+$ and the liquid water content in this work." has been added in lines 231-233.

4. Page 7, line 211-216, please give the locations of Guangzhou and Foshan in Figure S4.

**Response**: We have added the locations of Guangzhou and Foshan in Figure S4 as follows:

[Figure]

5. Page 8, line 239, in Figure 1a, the color of HM type particles is blue not purple?

**Response**: We did use purple color to represent the HM type particles in the manuscript but agreed that the purple color is not clearly distinguishable from red and light blue color. We have replaced purple color by orange color.

The original Figure 1(a) is as follows:

[Figure]

The revised Figure 1(a) with color change for HM type is as follows:

[Figure]

"In summer HM type particles (purple) and total oxalic acid particles exhibited similar diurnal patterns" is changed to "In summer HM type particles (orange color) and total oxalic acid particles exhibited similar diurnal patterns" in lines 269-270.

6. Page 10, section 3.2. Zn and Pb are typical tracers for vehicle emissions. The strong associations between heavy metals and oxalic acid in summer can be ascribed to the dominant emissions from vehicle exhausts. Oxalic acid can be produced via lots of formation pathways. Do you have direct evidence on the Fenton-like reaction? In the abstract section, authors stated that during summer ozone and oxalic acid-containing particles presented a similar temporal variation pattern, which indicates that photochemical reaction is important for oxalic acid production. Thus, as

for the above two formation mechanisms, which is more important for oxalic acid?

**Response**: Abundant secondary ions including m/z $-46[NO_2]^-$, $-62[NO_3]^-$, $-80[SO_3]^-$ and $-97[HSO_4]^-$ as well as $64[Zn]^+$ and $208[Pb]^+$ were found in HM type particles, suggesting that the HM type particles generally undergo an aging process after emitted from vehicle exhaust. This is supported by road-way tunnel measurements show a small contribution of direct emission from vehicle exhaust to the concentration of DCAs (Huang and Yu, 2007). In this work the sampling site is surrounded by farm land and villages, with no local industrial or traffic emissions. We suggest that HM type particles are mainly the result of regional transport after mixing and aging in the atmosphere.

In summer, oxalic acid-containing particles and ozone concentration followed a similar photochemical trend, and in-situ pH (2-4) was favorable for the Fenton-like reactions on HM type particles. We propose a photochemical formation pathway of oxalic acid in summer. Although the direct Fenton-like reactions were not observed in this work, the diurnal variation of ozone, HM particles and in-situ pH supported the possibility of oxalic acid production from the Fenton-like reactions. Based on above discussion we believe that photochemical reaction may be more important for oxalic acid production in this work.

We have also added the discussion about the contribution of traffic emission to oxalic acid formation in summer as follows:

"The influence from traffic emission was investigated through the diurnal variations of total EC type particles and $NO_2$ (Figure S7). The EC type particles increased from 12:00 to 21:00, which had same variation as total oxalic acid, but $NO_2$ followed the rush hour pattern with two peaks from 5:00 to 8:00 and from 18:00 to 21:00. Traffic emission is not expected to have a large contribution to oxalic acid in this study." has been added in lines 421-426.

[Figure]

Figure S7. The diurnal variations of temperature (T), RH, wind speed (WS), oxalic acid particles, total EC particles, the EC type oxalic acid-containing particles and ambient $NO_2$ concentrations from July 28 to August 1 in 2014.

**References:**

Huang, X.-F., and Yu, J. Z.: Is vehicle exhaust a significant primary source of oxalic acid in ambient aerosols?, Geophysical Research Letters, 34, L02808, 10.1029/2006gl028457, 2007.

Kawamura, K., Tachibana, E., Okuzawa, K., Aggarwal, S. G., Kanaya, Y., and Wang, Z. F.: High abundances of water-soluble dicarboxylic acids, ketocarboxylic acids and alpha-dicarbonyls in the mountaintop aerosols over the North China Plain during wheat burning season, Atmospheric Chemistry and Physics, 13, 8285-8302, 10.5194/acp-13-8285-2013, 2013.

Kundu, S., Kawamura, K., Andreae, T. W., Hoffer, A., and Andreae, M. O.: Molecular distributions of dicarboxylic acids, ketocarboxylic acids and alpha-dicarbonyls in biomass burning aerosols: implications for photochemical production and degradation in smoke layers, Atmospheric Chemistry and Physics, 10, 2209-2225, 10.5194/acp-10-2209-2010, 2010.

Li, L., Li, M., Huang, Z., Gao, W., Nian, H., Fu, Z., Gao, J., Chai, F., and Zhou, Z.: Ambient particle characterization by single particle aerosol mass spectrometry in an urban area of Beijing, Atmospheric Environment, 94, 323-331, 2014.

Stone, E. A., Hedman, C. J., Zhou, J. B., Mieritz, M., and Schauer, J. J.: Insights into the nature of secondary organic aerosol in Mexico City during the MILAGRO experiment 2006, Atmospheric Environment, 44, 312-319, 10.1016/j.atmosenv.2009.10.036, 2010.

Sullivan, R. C., and Prather, K. A.: Investigations of the diurnal cycle and mixing state of oxalic acid in individual particles in Asian aerosol outflow, Environmental Science Technology, 41, 8062-8069, 2007.

van Pinxteren, D., Neususs, C., and Herrmann, H.: On the abundance and source contributions of dicarboxylic acids in size-resolved aerosol particles at continental sites in central Europe, Atmospheric Chemistry and Physics, 14, 3913-3928, 10.5194/acp-14-3913-2014, 2014.

Yang, F., Chen, H., Wang, X., Yang, X., Du, J., and Chen, J.: Single particle mass spectrometry of oxalic acid in ambient aerosols in Shanghai: Mixing state and formation mechanism, Atmospheric Environment, 43, 3876-3882, 2009.

Zhang, G., Bi, X., Li, L., Chan, L. Y., Li, M., Wang, X., Sheng, G., Fu, J., and Zhou, Z.: Mixing state of individual submicron carbon-containing particles during spring and fall seasons in urban Guangzhou, China: a case study, Atmospheric Chemistry and Physics, 13, 4723-4735, 2013.

---

## Author Comment (AC2) · 6 Apr 2017

**Response to the comments of Anonymous Referee #2**

[Atmospheric Chemistry and Physics, MS ID: acp-2016-1081]
Title: Mixing state of oxalic acid containing particles in the rural area of Pearl River Delta, China: implication for seasonal formation mechanism of Secondary Organic Aerosol (SOA)

**General comments:**
The authors present a comprehensive analysis of the mixing state of oxalic acid in the Pearl River Delta area of China using a single particle aerosol mass spectrometer (SPAMS). The topic of the paper is important with regard to understanding formation pathways of secondary organic aerosol (SOA). The main findings are the following: oxalic acid containing particles accounted for <3% of total particles; in summer heavy metals containing particles were the largest group containing oxalic acid while in winter it was the biomass burning group; the majority of oxalic acid particles were internally mixed with sulfate and nitrate; the fraction of oxalic acid particles containing ammonium increases significantly in winter versus summer. A couple of interesting speculations are made about acid-catalyzed reactions and Fenton like reactions. The paper is written well. The title and abstract are mostly reflective of the contents of the paper. The results are quite interesting and I support publication after each of my comments below are adequately addressed.

**Response**: Thank you for your comments. These comments are all valuable and very helpful for revising and improving our paper, as well as the important guiding significance to our researches. We have studied these comments carefully and have made corrections. Our responses to the comments are itemized below.

Anything about our paper, please feel free to contact me at limei2007@163.com

Best regards!

Sincerely yours

Mei Li
April 6, 2017

**Specific comments and point by point responses:**

1. While the title of the paper and parts of the manuscript make reference to comparison of summer and winter and also mention the word 'seasonal', it is important to consider that the authors are only looking at two short-term periods spanning 2 weeks. I suggest a relaxation of words in parts of the paper that make it seem as though full seasonal coverage was obtained. At the minimum, 'seasonal' needs to be revised in my view for the title.

**Respons**e: As suggested by you and other reviewers, we have removed "seasonal" from the title, and the title "Mixing state of oxalic acid containing particles in the rural area of Pearl River Delta, China: implication for seasonal formation mechanism of Secondary Organic Aerosol (SOA)" has been changed to "Mixing state of oxalic acid containing particles in the rural area of Pearl River Delta, China: implications for the formation mechanism of oxalic acid".

2. Line 189: These percentages are quite interesting to me and they seem low based on how ubiquitous the literature suggests oxalic acid is in particles. Can the authors comment more about how these percentages compare to previous reports?

**Respons**e: Yang et al. (2009) has measured the oxalic acid-containing particles in the urban area of Shanghai by ATOFMS and found 15,789 oxalate-containing particles, accounting for 3.4% of the total collected particles. In this work 13,109 and 20,504 of oxalic acid-containing particles were obtained in summer and winter separately, accounting for 2.5% and 2.7% of the total detected particles. The abundance of oxalic acid-containing particles in this work was lower than the reported studies in the urban area of Shanghai (3.4%), which was possibly due to less anthropogenic precursors for oxalic acid at the rural sampling site in Heshan. Higher abundance of oxalic acid particles (1-40%) was observed in the much cleaner western Pacific Ocean by Sullivan et al. (2007), which corresponded to higher ambient concentration of DCAs (19±4.8%) in total particulate organic matter. From the reported studies in PRD (Yao et al. 2004; Ho et al. 2011), the abundance of DCAs was 1-3.5% in total organic matter, which was much lower than those in the western Pacific Ocean, leading to lower percentage of oxalic acid-containing particles in this work. We have added the comparison between this work and the study in Shanghai in the manuscript. Content of "The percentage of oxalic acid-containing particles in total particles in this work was comparable to the reported value in the urban area of Shanghai (3.4%) (Yang et al., 2009). However, these percentages are in general much lower than those reported in cleaner environments such as the western Pacific Ocean where oxalic acid was found in up to 1-40% of total particles due to little anthropogenic influences (Sullivan and Prather, 2007a)." has been added in lines 192-198.

3. Line 189-193: While the authors refer readers to the Supplement, it would be useful here to at least provide a little more detail as to how the categorization was done assuming that not all readers will go to the Supplement. The categories are very

important for the results, so some more discussion is warranted here as to how this was done.

**Response**: We have added the classification rules for oxalic acid particles in the manuscript and explain the differences among the EC, OC, ECOC type, and several literatures are cited in the manuscript to support these classification rules. "All oxalic acid particles are classified into eight types: elemental carbon (EC), organic carbon (OC), elemental and organic carbon (ECOC), biomass burning (BB), heavy metal (HM), secondary (Sec), sodium-potassium (NaK) and dust." has been changed to "The oxalic acid containing particles are classified into eight types in the following order: elemental carbon (EC), organic carbon (OC), elemental and organic carbon (ECOC), biomass burning (BB), heavy metal (HM), secondary (Sec), sodium-potassium (NaK) and dust. Different type particles are identified according to characteristic ion markers and dominant chemical species (Table S1): (1) particles containing abundant carbon clusters like $\pm12[C]^{+/-}$, $\pm24[C_2]^{+/-}$, $\pm36[C_3]^{+/-}$ with relative peak area more than 0.5% are classified as EC type, (2) any remaining particles containing abundant signals of $27[C_2H_3]^+$, $43[C_2H_3O]^+$ and hydrocarbon clusters with relative peak area more than 0.5% are classified as OC type, (3) any remaining particles containing signals of $\pm12[C]^{+/-}$, $\pm24[C_2]^{+/-}$, $37[C_3H]^+$ and $43[C_2H_3O]^+$ with relative peak area more than 0.5% are classified as ECOC type, (4) any remaining particles containing abundant signals of $39[K]^+$ (peak area>1500) with relative peak area of $-59[C_2H_3O_2]^-$ and $-73[C_3H_5O_2]^-$ simultaneously more than 0.5% are classified as BB type, (5) any remaining particles containing signals of $55[Mn]^+$, $56[Fe]^+$, $63/65[Cu]^+$, $64[Zn]^+$ and $208[Pb]^+$ with relative peak area more than 0.5% are classified as HM type, (6) any remaining particles containing abundant signals of $18[NH_4]^+$ (peak area>50), $-62[NO_3]^-$ (peak area>100) and $-97[HSO_4]^-$ (peak area>100) are classified as Sec type, (7) any remaining particles containing abundant signals of $23[Na]^+$ (peak area>1500) and related species are classified as NaK type, (8) any remaining particles containing signals of $40[Ca]^+$, $56[CaO]^+$ and related species are classified as dust type. The rules for oxalic acid particles classification in the current work have been reported in previous studies (Sullivan and Prather, 2007b;Yang et al., 2009;Zhang et al., 2013;Li et al., 2014)." in lines 199-221.

4. Line 289-291: Could the lower mixing height in the winter have contributed in some way to this finding (i.e. more stagnant aerosol and perhaps more aged)?

**Response**: The wind speed was from 0.3 to 4.0 m s$^{-1}$ with an average of 1.6 m s$^{-1}$ during the sampling period in winter, which indicated a rather stagnant atmospheric condition. The stagnant atmospheric condition in winter was favorable for the aging process of aerosols and might have contributed to the broader size distribution of oxalic acid particles. Unfortunately the height of boundary layer was not available in this work. Nevertheless, the relative abundance of oxalic acid particles had no obvious increase at night and the lower mixing height may not be an important factor during the formation process of oxalic acid in the winter. We will further examine their relationships in our next study.

5. Line 337-338: The wording here in this sentence and the general paragraph appear to be too strong in my view since the authors did not unambiguously prove that photo- Fenton reactions are even occurring. Aren't these just speculations? I suggest to use less strong language and to differentiate better between proved findings and speculations.

**Response**: Indeed these speculations and proposed results should be addressed in a proper expression. "Although •OH production from Fenton reactions can both occur under dark and light radiation conditions, only photo-Fenton reactions had significant contribution to the enhancement of oxalic acid particles in the current work." has been changed to "Although •OH production from Fenton reactions can both occur under dark and light radiation conditions, photo-Fenton reactions may have more contribution to the enhancement of oxalic acid particles in the current work." in lines 369-371.

6. Line 329-359: have the authors considered all other factors that could affect the diurnal behavior of oxalic acid particles, such as meteorological factors or different emissions types during the day. Can the authors comment on how the profiles of other species look in Figure 7 if they were plotted such as sulfate, EC, nitrate? Is rush hour traffic influential at all in any of the discrepancies between the peak of ozone and the other parameters currently shown in Figure 7? While these other factors may not be important, it is still important to mention that various other potential factors were considered.

**Response**: The diurnal variations of meteorological factors such as temperature, RH and wind speed are shown in Figure S7.

[Figure]

Figure S7. The diurnal variations of temperature (T), RH, wind speed (WS), oxalic acid particles, total EC particles, the EC type oxalic acid-containing particles and ambient $NO_2$ concentrations from July 28 to August 1 in 2014.

The high temperature between 9:00 and 19:00 was favorable to the secondary

processing of organic precursors. The wind speed was low during the whole day, especially between 9:00 and 18:00, which provided a stagnant environment for the increase in oxalic acid after produced from photochemical process. The impacts of RH and traffic emissions on the formation of oxalic acid have been illustrated Figure S5 and S7, respectively. The influence from traffic emission was investigated through the diurnal variations of total EC type particles and $NO_2$ (Figure S7). The EC type particles increased from 12:00 to 21:00, which had same variation as total oxalic acid, but $NO_2$ followed the rush hour pattern with two peaks from 5:00 to 8:00 and from 18:00 to 21:00. Traffic emission is not expected to have a large contribution to oxalic acid in this study. Oxalic acid was found to be internally mixed with sulfate and nitrate with mixing ratio of 93% and 94% in summer, and the sulfate and nitrate type oxalic acid particles were not classified in this work, so the diurnal patterns of sulfate and nitrate are not presented in Figure S7.

We have added the related discussion in the manuscript. "The diurnal patterns of temperature, wind speed are presented in Figure S7. The high temperature between 9:00 and 19:00 was favorable to the secondary processing of organic precursors. The wind speed was low during the whole day, especially between 9:00 and 18:00, which provided a stagnant environment for the increase in oxalic acid produced from photochemical process. The influence from traffic emission was investigated through the diurnal variations of total EC type particles and $NO_2$ (Figure S7). The EC type particles increased from 12:00 to 21:00, which had same variation as total oxalic acid, but $NO_2$ followed the rush hour pattern with two peaks from 5:00 to 8:00 and from 18:00 to 21:00. Traffic emission is not expected to have a large contribution to oxalic acid in this study." has been added in lines 417-426 in the manuscript.

7. Section 3.3: the discussion and analysis surrounding the acid-catalyzed hypothesis is too thin in my view. Were such relationships not observed in the summer, and if not, then why? Could another reasonable explanation be that that precursors of oxalic acid and the organosulfate species are co-emitted? To strengthen this conclusion, the discussion and analysis needs to be more convincing with also more discussion of other relevant words using field data to point to this mechanism of acid-catalyzed formation of SOA.

**Response**: In summer few organosulfate containing particles were observed and hence we have only discussed the relationship between the oxalic acid and organosulfate containing particles in winter. The temporal trend of organosulfate-containing oxalic acid particles in winter has been added in Figure S8, which exhibited a similar pattern as the total oxalic acid particles. The percentage of organosulfate-containing oxalic acid particles in total oxalic acid particles ranged from 0 to 16.4% with the highest ratio observed on February 8. The linear regression between oxalic acid particles and organosulfate particles in Figure 9b has been replaced by the correlation between organosulfate-containing oxalic acid particles and total oxalic acid particles, and the robust correlation ($r^2$=0.81) between them supports a possible production of oxalic acid from acid-catalyzed reactions.

We agree with more evidence and discussion are very helpful to support the

proposed mechanism connecting oxalic acid formation and aqueous phase chemistry at acidic conditions, and following discussions as well as a new Figure 8 have been added in lines 428-455:

"Despite lower $O_3$ concentrations and photochemical activity in winter, oxalic acid particles were still prevalent in carbonaceous particles, especially BB type particles. While oxalic acid was found to be internally mixed with sulfate and nitrate both in summer and winter, the nitric acid was only observed in oxalic acid particles in winter, indicating a strongly acidic nature of oxalic acid particles in winter. Considering a possible connection of oxalic acid production with the acidic environment, the temporal concentrations of oxalic acid, sulfate and nitrate were investigated through their peak areas in the carbonaceous type oxalic acid particles including EC, OC, ECOC and BB type in Figure 8. The peaks of m/z -62[$NO_3$]$^-$ and -97[$HSO_4$]$^-$ represent nitrate and sulfate, respectively. Nitrate, sulfate and oxalic acid showed very similar variation patterns in winter, indicating a close connection of their co-existence. Although nitric acid was found in the oxalic acid particles, the acidity of the oxalic acid particles was not estimated since the real-time concentration of inorganic ions was not available during the sampling period in winter. Instead the relative acidity ratio ($R_{ra}$), defined as the ratio of total peak areas of nitrate and sulfate to the peak area of ammonium (m/z 18[$NH_4$]$^+$), was used (Denkenberger et al., 2007;Pratt et al., 2009). The $R_{ra}$ of carbonaceous type oxalic acid particles ranged from 7 to 114 with an average value of 25 (Figure 8), indicating an intensely acidic environment of carbonaceous type oxalic acid particles in winter. Several studies have reported potential production of oxalic acid from acid-catalyzed aqueous phase reactions in aerosols (Carlton et al., 2006;Carlton et al., 2007;Tan et al., 2009). In this work the acidic environment of the carbonaceous type oxalic acid particles and similar variation patterns among oxalic acid, sulfate and nitrate may suggest a relationship between the degradation of organic precursors and the acidic chemical process. However, the temporal change of $R_{ra}$ did not follow a similar trend as the peak area of oxalic acid in most particles, possibly due to the multi-step formation of oxalic acid influenced by many factors such as precursors, liquid water content and ion strength (Carlton et al., 2007;Cheng et al., 2013;Cheng et al., 2015)."

[Figure]

Figure 8. The temporal variations of peak area of nitrate, sulfate and oxalic acid, and the relative acidity ratio ($R_{ra}$) in carbonaceous type oxalic acid particles in winter.

The related discussions about Figure 9 and Figure S8 have been revised: "The organosulfate derived from glyoxal requires acidic aqueous environment of particles, and herein is used as an indicator of acid-catalyzed ageing process of organic compounds. The temporal variation of organosulfate (m/z=-155) containing particles during the entire sampling period in Heshan, China is shown in Figure S6. During the episode, oxalic acid particles had moderate linear correlation with organosulfate particles (Figure 9b)." has been changed to "The formation of organosulfates from glyoxal requires an acidic aqueous environment, which can be used as an indicator of acid-catalyzed ageing process of organic compounds. The temporal trend of organosulfate-containing oxalic acid particles in winter is shown in Figure S8, which exhibited a similar pattern as the total oxalic acid particles during the whole sampling period in winter. The percentage of organosulfate-containing oxalic acid particles in total oxalic acid particles ranged from 0 to 16.4% with the highest ratio observed in the episode (February 8). The linear regression between total oxalic acid particles and organosulfate-containing oxalic acid particles in the episode is exhibited in Figure 9b, and the robust correlation ($r^2$=0.81) between them suggests that oxalic acid and organosulfate may share similar formation process." in lines 476-486.

[Figure]

Figure 9. The comprehensive study of oxalic acid particles increase on Feb 8, 2015: (a) The digitized positive and negative ion mass spectrum of oxalic acid particles during the episode; (b) Linear regression between total oxalic acid particles and organosulfate-containing oxalic acid particles (m/z -155).

[Figure]

Figure S8. Temporal variation of organosulfate (m/z=-155) containing particles in total particles and in oxalic acid particles in Heshan, China.

"The temporal trend of organosulfate-containing oxalic acid particles in winter is also shown in Figure S8, which exhibited a similar pattern as the total oxalic acid particles. The percentage of organosulfate-containing oxalic acid particles in total oxalic acid particles ranged from 0 to 16.4% with the highest ratio observed on February 8. " has been added in the supplement material.

8. Line 107-110: This line needs a revision because it is not entirely fair. It should be noted somewhere around this section that quite a bit of work has been done with fast time resolution on aircraft to address the issue of meteorological uncertainty and temporal resolution limitations. These various studies have discussed the formation pathways leading to oxalic acid with detailed in-cloud and out-of-cloud measurements:

Wonaschuetz, A., et al. (2012). Aerosol and gas re-distribution by shallow cumulus clouds: an investigation using airborne measurements, J. Geophys. Res., 117, D17202, doi:10.1029/2012JD018089.

Sorooshian, A., et al. (2006). , J. Geophys. Res. 111, D23S45, doi:10.1029/2005JD006880.

Sorooshian, A., et al. (2007). Particulate organic acids and overall water-soluble aerosol composition measurements from the 2006 Gulf of Mexico Atmospheric Composition and Climate Study (GoMACCS), J. Geophys. Res., 112, D13201, doi:10.1029/2007JD008537.

**Response**: We have revised these sentences, and these references related to in-cloud and aqueous phase formation of oxalic acid have also been cited in the manuscript. "Several studies have found a tight correlation between oxalic acid and sulfate in ambient particles, implying that aqueous chemistry leads to the formation of oxalic acid in aerosols and cloud droplets (Yao et al., 2002;Yao et al., 2003;Yu et al., 2005;Sorooshian et al., 2007a;Miyazaki et al., 2009)." has been changed to "A number of ground based and airborne field studies have found a tight correlation between oxalic acid and sulfate in ambient particles and cloud droplets, relating

aqueous phase chemistry to the formation of oxalic acid in aerosols and cloud droplets (Yao et al., 2002;Yao et al., 2003;Yu et al., 2005;Sorooshian et al., 2006;Sorooshian et al., 2007a;Sorooshian et al., 2007b;Miyazaki et al., 2009;Wonaschuetz et al., 2012;Wang et al., 2016)." in line 99-104. "However, the exact formation pathways of oxalic acid in ambient particles are still unknown due to the complexity of meteorological condition and the temporal resolution limitations of conventional filter sampling studies and bulk chemical analysis." has been changed to "However, the detailed formation mechanisms of oxalic acid from photochemistry and aqueous phase chemistry in ambient aerosols are still not comprehensively understood and need to be further studied." in lines 111-113.

9. Figures: Font size needs to increase in many of the figures for labels.
**Response**: We have checked all the Figures and Tables and revised the font size in most figures in the manuscript and the supplement material.

**References:**

[revised manuscript text omitted]

---

## Author Comment (AC3) · 6 Apr 2017

**Response to the comments of Anonymous Referee #3**

[Atmospheric Chemistry and Physics, MS ID: acp-2016-1081]
Title: Mixing state of oxalic acid containing particles in the rural area of Pearl River Delta, China: implication for seasonal formation mechanism of Secondary Organic Aerosol (SOA)

**General comments:**

The authors present results measurements of ambient aerosol during two periods (summer vs winter) in the Pearl River Delta. Most of the discussion focuses on oxalate loadings and its mixing state. Based on correlations with metal-containing particles, the authors conclude that iron has an important role in OH formation and therefore in oxalate formation. While generally the manuscript is well written, the discussion of the chemical mechanism is very weak and hand waving. Without running a detailed multiphase model, such conclusions cannot be drawn with certainty. Therefore, I suggest removing the discussion of the role of iron and the chemical mechanism in general. After my additional comments below are addressed, the manuscript may be suitable for publication.

**Response**: Thank you for your comments and suggests. The substantial internal mixing of oxalic acid with sulfate and nitrate in summer reflects the secondary nature of oxalic acid particles. Together with the abundant heavy metal type oxalic acid particles, it suggests a strong connection between the production of some oxalic acid and the participation of transition metals like iron and copper. The photochemical formation process of oxalic acid in summer is proposed to explain the obvious photochemical diurnal pattern of HM type oxalic acid particles. Indeed, it would be more convincing if we run the multiphase model to compare the results of proposed explanation and the observation in the field measurements. However, we are not equipped to run such model simulations. Thus, we have toned down about the discussions of the production of oxalic acid in HM type particles as a plausible explanation instead of a proposed formation mechanism.

We have answered all your comments and questions through point by point response to illustrate the photochemical formation process of oxalic acid in this work. We appreciate these valuable and helpful comments to our work and enable it to meet the high quality of the journal Atmos. Chem. Phys. Our responses to all the comments are itemized below.

Anything about our paper, please feel free to contact me at limei2007@163.com

Best regards!

Sincerely yours
Mei Li
April 6, 2017

**Specific comments and point by point responses:**

Major comments

1. The suggested chemical mechanism is oversimplified and contains several misconceptions and/or omissions:

a) Generally, it is assumed that direct OH uptake is the main source of OH in the aqueous phase [e.g., Ervens et al., 2003; Herrmann, 2003; Tilgner et al., 2013]. Therefore higher OH (gas) concentration will lead to higher OH concentration in the aqueous phase. Higher OH (aq) concentration will also lead to more oxidation of oxalate and therefore less SOA.

**Response**: We thank the reviewer's comments. The main sources of •OH in the aqueous phase contain both direct uptake from the gas phase and chemical sources in the aqueous phase such as Fenton reactions and photolysis of $H_2O_2$ and nitrate (Ervens et al., 2014;Ervens, 2015;Gligorovski et al., 2015;Herrmann et al., 2015). Although the concentrations of $H_2O_2$ and •OH in gas and aqueous phases are not available in this work, considering much higher Henry's law constant of $H_2O_2$ ($K_H=8.3\times10^4$ M atm$^{-1}$) than •OH ($K_H=30$ M atm$^{-1}$) (Hanson et al., 1992;O'Sullivan et al., 1996) and abundant fraction of transition metals in the oxalic acid particles in this work, aqueous phase chemistry likely contributes substantially to the source of •OH in the aqueous phase according to high efficiency of •OH production from Fenton type reactions (Gligorovski et al., 2015). The aqueous phase oxidation of glyoxal and methylglyoxal is the main source of oxalic acid production, and their reaction rates with •OH ($1.1\times10^9$ mole$^{-1}$ s$^{-1}$) is one order of magnitude higher than oxalic acid with •OH ($1.9\times10^8$ mole$^{-1}$ s$^{-1}$) as previously reported (Herrmann, 2003;Myriokefalitakis et al., 2011). Thus, the high •OH (aq) concentration would lead to the enrichment of oxalic acid in aqueous phase.

In order to clarify these points, we have changed "While the partition of •OH from gas to aqueous phase is limited by its low Henry's law constant ($K_{H,OH}=30$ M atm$^{-1}$) and short lifetime of •OH in the gas phase (Hanson et al., 1992), the main sources of aqueous •OH are from the photolysis of $H_2O_2$, $NO_3^-$, $NO_2^-$, and chromophoric dissolved organic matter (CDOM) (Yu et al., 2014;Badali et al., 2015;Gligorovski et al., 2015;Tong et al., 2016). Among these sources the photolysis of $H_2O_2$ through Fenton reactions involving the catalysis of transition metal ions like $Fe^{2+/3+}$, $Cu^{+/2+}$ and $Mn^{2+/3+}$ is an efficient source of •OH (Deguillaume et al., 2005;Herrmann et al., 2005;Ervens et al., 2014)." to "The main sources of •OH in the aqueous phase contain both direct uptake from the gas phase and the chemical sources in the aqueous phase such as Fenton type reactions and photolysis of $H_2O_2$, $NO_3^-$, $NO_2^-$, and chromophoric dissolved organic matter (CDOM) (Yu et al., 2014;Badali et al., 2015;Ervens, 2015;Gligorovski et al., 2015;Herrmann et al., 2015;Tong et al., 2016). Considering the low Henry's law constant of •OH ($K_{H,OH}=30$ M atm$^{-1}$) (Hanson et al., 1992) and abundant fraction of transition metal ions in the oxalic acid particles, the photolysis of $H_2O_2$ through Fenton reactions involving the catalysis of transition metal ions like $Fe^{2+/3+}$, $Cu^{+/2+}$ and $Mn^{2+/3+}$ likely contributes substantially to the source of •OH in the aqueous phase in this work (Deguillaume et al., 2005;Herrmann et al., 2005;Ervens et al., 2014)." in lines 345-355.

b) Higher iron concentration might lead to more OH. However, more importantly is the effect of the loss of oxalate due to the photolysis of the iron-oxalato complex. While this reaction is mentioned in the manuscript, its predominating role in oxalate loss [Sorooshian et al., 2013] is not discussed in a balanced way.

**Response**: The degradation of oxalic acid from the complex with iron has been discussed in the manuscript to explain the diurnal change of oxalic acid particles. Indeed, it would be more specific to evaluate the net production of oxalic acid from photochemical process if we could calculate the production and loss of oxalic acid. Unfortunately, we have no real time data about the ambient concentrations of oxalic acid and iron, so the exact amount of oxalic acid loss from the photolysis of the iron complexes cannot be obtained for now.

c) At low pH, it can be expected that reaction rates are lower since in general the undissociated acids (glyoxylic, glycolic) react more slowly than their dissociated counterparts. Oxalate has a very low pKa (1.23) so that even at low pH a substantial fraction is still present as oxalate. Could changes in pH and therefore reaction rates explain some of the temporal trends?

**Response**: Indeed, the reaction rate of glyoxylate with •OH ($k=2.8\times10^9$ mole$^{-1}$ s$^{-1}$, pH=8) is one order of magnitude higher than glyoxylic acid ($k=3.6\times10^8$ mole$^{-1}$ s$^{-1}$, pH=1), so the aqueous phase with high pH value is more favorable for the production of oxalic acid from the oxidation of precursors. The SPAMS can only detect the ion peak of oxalate ion (m/z -89) in single particles, so the percentage of oxalate in total oxalic acid cannot tell from the data of SPAMS. We believe the diurnal change of pH also has influence on the oxidative process of organic precursors in addition to provide a proper environment for Fenton like reactions. We have added the related discussion in the manuscript. "In addition to the contribution from Fenton reactions after 12:00, the precursors of oxalic acid such as glyoxylic acid have higher reaction rate with •OH at high pH based on previous studies(Ervens et al., 2003;Herrmann, 2003;Cheng et al., 2015), thus the increase of pH not only enhances •OH production from photo-Fenton reactions, but also promotes the oxidation process of the precursors of oxalic acid by •OH." has been added in lines 392-397.

d) At very low pH (< 1.23), it is expected that oxalic acid is present in undissociated form and therefore not able to make salts or complexes that 'trap' it in the particle phase. This fact contradicts the trend of increased oxalate concentrations at low pH. This should be discussed.

**Response**: Oxalic acid is predominantly enriched in particle phase due to its low vapor pressure ($3.5\times10^{-5}$ Torr) (Prenni et al., 2001) and high water solubility. Several studies have indicated that more than 70% of oxalic acid could exist in particle phase (Limbeck et al., 2001;Mochida et al., 2003a;Mochida et al., 2003b). The salt and complex formation of oxalate with ammonium, potassium, sodium and metal ions

have been considered as the main behavior of oxalic acid in the particle phase (Yao et al., 2002;Moffet et al., 2008;Furukawa and Takahashi, 2011), and a higher pH condition would be more favorable for the salt and complex formation. Due to the abundant signals of potassium, sodium and metal ions in this work, oxalic acid can react with these anions and stay in the particle phase. Besides, the oxalic acid particles and in-situ pH both exhibited an increase in the afternoon, which was in accordance with the announcement of more oxalate at higher pH.

e) The proposed mechanism is by no means new or detailed (l. 360). It does not include any sinks of oxalate, nor complex formation. It is one possible formation mechanism of oxalic acid from glyoxal. The generalization to dicarbonyls and aldehydes is not correct since only small compounds (C2) will follow the suggested reaction pathways.

**Response**: The photochemical production of oxalic acid has been studied in many field researches due to a similar diurnal pattern with $O_3$ (Kawamura and Ikushima, 1993;Kawamura and Yasui, 2005;Aggarwal and Kawamura, 2008;Miyazaki et al., 2009), but the detailed photochemical formation process of oxalic acid has not been comprehensively discussed. While the oxidation of glyoxal and glyoxylic acid has been proposed and discussed in many field, laboratory and model studies (Myriokefalitakis et al., 2011;Wang et al., 2012;Kawamura et al., 2013;Ervens et al., 2014;Wang et al., 2014), the influential factors of their oxidation reactions with •OH still needs to be ascertained from the field studies. Based on a comprehensive discussion of the mixing state of oxalic acid with secondary ions and transition metals, we propose a plausible explanation to connect among the diurnal pattern of $O_3$, oxalic acid, iron and in-situ pH. (Sorooshian et al., 2013;Zhou et al., 2015). However, without the direct measurement of the concentrations of •OH, iron ions, and oxalic acid, the proposed photochemical formation mechanism of oxalic acid in the original manuscript is difficult to be confirmed. Thus, we have changed the expression about the oxalic acid production in HM type particles into a plausible explanation instead of a proposed formation mechanism, and have softened our tone in the discussions. The schematic diagram to explain the formation process of oxalic acid in the HM type particles has been moved to the supplement material in Figure S6. The related discussion has been revised as follows:

In the abstract "Furthermore, favorable in-situ pH (2-4) conditions were observed, which promote Fenton like reactions for efficient production of •OH in HM type particles. A mechanism in which products of photochemical oxidation of VOCs partitioned into the aqueous phase of HM particles, followed by multistep oxidation of •OH through Fenton like reactions to form oxalic acid is proposed." has been changed to "The favorable in-situ pH (2-4) and the dominance of transition metal ions in oxalic acid particles can be plausibly explained by the enhanced production of •OH from Fenton like reaction, which can promote the oxalic acid production from the oxidation of precursors by •OH in HM type particles." in lines 55-58.

"Based on above discussions, detailed mechanism for oxalic acid formation in acidic aqueous phase of particles is proposed for our field observations (Figure 8)."

has been changed to "Based on above discussions of the mixing state of oxalic acid with secondary ions and transition metals, a plausible explanation to the formation process of oxalic acid in the HM type oxalic acid particles is proposed (Figure S6)." in lines 398-400.

In the conclusion "The favorable in-situ pH and the dominance of transition metal ions in oxalic acid particles suggests an enhanced production of •OH from Fenton like reactions. A mechanism involving the photochemical production of VOCs via efficient aqueous phase reactions with enhanced •OH to oxalic acid was proposed." has been changed to "Furthermore, suitable in-situ pH is favorable for Fenton like reactions to produce •OH in HM type particles, and might promote the oxalic acid production from the oxidation of precursors by •OH in HM type particles." in lines 504-506.

To address the referee's concern, we have revised the description of dicarbonyls and aldehydes in Figure S6. The "dicarbonyls and aldehydes" has been changed to "Low molecular weight dicarbonyls (e.g. glyoxal)", and "Hydrated dicarbonyls and aldehydes" has been changed to "Hydrated dicarbonyls (e.g. glyoxal)" in Figure S6:

[Figure]

Figure S6. A schematic diagram to explain the formation process of oxalic acid in the HM type particles: the red steps are enhanced by photochemical activities in the current study.

2. The number fraction of oxalate containing particles seems very low. Is this comparable to other measurements? What was the mass fraction of oxalate (a) in the particles and (b) related to the total aerosol loading?

**Response**: Yang et al. (2009) has measured the oxalic acid-containing particles in the urban area of Shanghai by ATOFMS and found 15,789 oxalate-containing particles, accounting for 3.4% of the total collected particles. In this work 13,109 and 20,504 of oxalic acid-containing particles were obtained in summer and winter

separately, accounting for 2.5% and 2.7% of the total detected particles. The abundance of oxalic acid-containing particles in this work was lower than the reported studies in the urban area of Shanghai (3.4%), which was possibly due to less anthropogenic precursors for oxalic acid at the rural sampling site in Heshan. Higher abundance of oxalic acid particles (1-40%) was observed in the much cleaner western Pacific Ocean by Sullivan et al. (2007), which corresponded to higher ambient concentration of DCAs (19±4.8%) in total particulate organic matter. From the reported studies in PRD (Yao et al. 2004; Ho et al. 2011), the abundance of DCAs was 1-3.5% in total organic matter, which was much lower than those in the western Pacific Ocean, leading to lower percentage of oxalic acid-containing particles in this work. We have added the comparison between this work and the study in Shanghai in the manuscript. "The percentage of oxalic acid-containing particles in total particles in this work was comparable to the reported value in the urban area of Shanghai (3.4%) (Yang et al., 2009). However, these percentages are in general much lower than those reported in cleaner environments such as the western Pacific Ocean where oxalic acid was found in up to 1-40% of total particles due to little anthropogenic influences (Sullivan and Prather, 2007)." is added in lines 192-198.

For the second question, the relative fraction of oxalic acid particles in total detected particles was accounting for 2.5% and 2.7% of the total detected particles in summer and winter (Figure 1a). However, because we didn't collect filter samples during the sampling period, we cannot calculate the relative mass fraction of oxalic acid in ambient $PM_{2.5}$.

3. Was all iron in the particles in form of soluble iron, i.e. available for reaction?

**Response**: Indeed, it would be more convincing to discuss the role of Fenton reactions in the formation of oxalic acid particles if we could obtain the concentration of $Fe^{2+/3+}$ ions. However, we didn't collect filter samples during the sampling period, so the concentration of $Fe^{2+/3+}$ ions were not measured. Besides, the SPAMS could only detect the peak area of iron element, and the different valence state of iron could not be distinguished. Therefore, the discussions between Fenton reactions and oxalic acid production have been revised to a plausible explanation instead of a proposed formation mechanism.

4. Oxalate and the other DCAs usually represent only a very small fraction of the total organic aerosol mass. Therefore the title is misleading as it talks about SOA in general.

**Response**: Based on the comments from you and other reviews, the title "Mixing state of oxalic acid containing particles in the rural area of Pearl River Delta, China: implication for seasonal formation mechanism of Secondary Organic Aerosol (SOA)" has been changed to "Mixing state of oxalic acid containing particles in the rural area of Pearl River Delta, China: implications for the formation mechanism of oxalic

acid".

5. It seems based on Figure S5, that RH was always < 100% (except a very brief period). Therefore, the discussed aqueous chemistry will have to take place in aqueous aerosol. There are many studies that have discussed different reaction pathways in aqueous aerosol vs cloud [e.g., Tan et al., 2009; Lim et al., 2010] with less efficient oxalate formation in the former. In addition, it seems likely that iron ions might be less dissolved in the rather highly concentrated aqueous aerosol solutions. All discussion is about chemistry as it happens in cloud droplets. These two regimes should be differentiated.

**Response**: While the aqueous phase formation of oxalic acid from organic precursors has been found both in cloud droplets and aerosols water, the more efficient production in cloud droplets than aerosols water has been reported by several studies (Tan et al., 2009;Lim et al., 2010;Myriokefalitakis et al., 2011). In this work, the well internally mixing state between oxalic acid and sulfate in summer and winter suggests a common production route of oxalic acid and sulfate, likely aqueous phase reactions. The strong photochemical pattern of oxalic acid particles with $O_3$ under the condition of RH below 100% implies a less influence from cloud-processing of oxalic acid particles. Thus we speculate the oxalic acid formation process in HM type particles is related to aqueous phase reactions in aerosols, and this point has been stated in the manuscript: "Based on above discussions of the mixing state of oxalic acid with secondary ions and transition metals, a plausible explanation to the formation process of oxalic acid in the HM type oxalic acid particles is proposed (Figure S6). In summer strong photochemical activity and high $O_3$ concentrations in the afternoon lead to more production of reactive radicals such as •OH and $HO_2{}^{\bullet}$, which promote the oxidation of VOCs to dicarbonyls and aldehydes (e.g. glyoxal and methylglyoxal), followed by a subsequent partitioning into the aqueous phase of particles" in lines 398-404.

Although the reviewer has pointed out that iron ions might be less dissolved in the rather highly concentrated aqueous aerosol solutions, the low in-situ pH of aerosols is also favorable for the dissolved of iron. Because we didn't collect filter samples during the sampling period, the concentration of $Fe^{2+/3+}$ ions were not measured, so the ratio of $Fe^{2+/3+}$ ions to total iron element could not be evaluated for now, and we will further discuss this issue in our next study.

6. While briefly discussed, it is not clear to what extent different air masses cause different oxalate levels. How much of the measured oxalate is background material? Did other meteorological conditions affect the concentrations such as changes in boundary layer?

**Response**: The different air masses arriving in sampling site in summer and winter had substantial impact on the amount of organic precursors from

anthropogenic emissions. More carbonaceous signals were found in the oxalic acid particles in winter since oxalic acid is mainly derived from secondary oxidation of VOCs and subsequent intermediates. However, it is still difficult to quantify exactly the increase of oxalic acid concentration in winter based on the SPAMS data. The sampling site in this work is a rural site and the aerosols are influenced by the regional transport in PRD. Although we cannot obtain the exact background level of oxalic acid, the lowest percentage of oxalic acid particles in total detected particles is used to evaluate the approximate background level of oxalic acid particles. The lowest hourly percentage of oxalic acid particles in total detected particles were 0.1% and 0.5 % in summer and winter, which were much lower than the average value in summer (2.3%) and winter (2.8%), suggesting a small impact of the background distribution to the measurements of oxalic acid particles.

The wind speed was from 0.3 to 4.0 m s$^{-1}$ with an average of 1.6 m s$^{-1}$ during the sampling period in winter, which indicated a rather stagnant atmospheric condition. The stagnant atmospheric condition in winter was favorable for the aging process of aerosols and might have contributed to the broader size distribution of oxalic acid particles. Unfortunately the height of boundary layer was not available in this work. Nevertheless, the relative abundance of oxalic acid particles had no obvious increase at night and the lower mixing height may not be an important factor during the formation process of oxalic acid in the winter. We will further examine their relationships in our next study.

The diurnal variations of meteorological factors such as temperature, RH and wind speed have been discussed in Figure S7.

[Figure]

Figure S7. The diurnal variations of temperature (T), RH, wind speed (WS), oxalic acid particles, total EC particles, the EC type oxalic acid-containing particles and ambient NO$_2$ concentrations from July 28 to August 1 in 2014.

We have added the related discussion in the manuscript. "The diurnal patterns of temperature, wind speed are presented in Figure S7. The high temperature between 9:00 and 19:00 was favorable to the secondary processing of organic precursors. The

wind speed was low during the whole day, especially between 9:00 and 18:00, which provided a stagnant environment for the increase in oxalic acid produced from photochemical process. The influence from traffic emission was investigated through the diurnal variations of total EC type particles and $NO_2$ (Figure S7). The EC type particles increased from 12:00 to 21:00, which had same variation as total oxalic acid, but $NO_2$ followed the rush hour pattern with two peaks from 5:00 to 8:00 and from 18:00 to 21:00. Traffic emission is not expected to have a large contribution to oxalic acid in this study." has been added in lines 417-426 in the manuscript.

7. I am not sure what Figure S6 is really showing. Does it show a correlation of organosulfur particles and oxalate or does it simply show that more particles cause higher concentrations of 'everything'? How about the mixing state of organosulfur compounds and oxalic acid particles? The fact that they are in the same particle class, does not necessarily mean that they are internally mixed and therefore their formation pathways are related.

**Response**: The original Figure S6 (now Figure S8) presented the temporal variation of total organosulfate (m/z=-155) containing particles during whole sampling periods in Heshan, China. The temporal trend of organosulfate-containing oxalic acid particles in winter has been added in Figure S8, which exhibited a similar pattern as the total oxalic acid particles. The percentage of organosulfate-containing oxalic acid particles in total oxalic acid particles ranged from 0 to 16.4% with the highest ratio observed on February 8. The linear regression between oxalic acid particles and organosulfate particles in Figure 9b has been replaced by the correlation between organosulfate-containing oxalic acid particles and total oxalic acid particles, and the robust correlation ($r^2$=0.81) between them supports a possible production of oxalic acid from acid-catalyzed reactions.

We also believe more evidence and discussion are needed to support the connection between oxalic acid formation process and acidic aqueous phase chemistry. So the related discussions have added as follows:

"Despite lower $O_3$ concentrations and photochemical activity in winter, oxalic acid particles were still prevalent in carbonaceous particles, especially BB type particles. While oxalic acid was found to be internally mixed with sulfate and nitrate both in summer and winter, the nitric acid was only observed in oxalic acid particles in winter, indicating a strongly acidic nature of oxalic acid particles in winter. Considering a possible connection of oxalic acid production with the acidic environment, the temporal concentrations of oxalic acid, sulfate and nitrate were investigated through their peak areas in the carbonaceous type oxalic acid particles including EC, OC, ECOC and BB type in Figure 8. The peaks of m/z -62[$NO_3$]$^-$ and -97[$HSO_4$]$^-$ represent nitrate and sulfate, respectively. Nitrate, sulfate and oxalic acid showed very similar variation patterns in winter, indicating a close connection of their co-existence. Although nitric acid was found in the oxalic acid particles, the acidity of the oxalic acid particles was not estimated since the real-time concentration of inorganic ions was not available during the sampling period in winter. Instead the

relative acidity ratio ($R_{ra}$), defined as the ratio of total peak areas of nitrate and sulfate to the peak area of ammonium (m/z 18[$NH_4$]$^+$), was used (Denkenberger et al., 2007;Pratt et al., 2009). The $R_{ra}$ of carbonaceous type oxalic acid particles ranged from 7 to 114 with an average value of 25 (Figure 8), indicating an intensely acidic environment of carbonaceous type oxalic acid particles in winter. Several studies have reported potential production of oxalic acid from acid-catalyzed aqueous phase reactions in aerosols (Carlton et al., 2006;Carlton et al., 2007;Tan et al., 2009). In this work the acidic environment of the carbonaceous type oxalic acid particles and similar variation patterns among oxalic acid, sulfate and nitrate may suggest a relationship between the degradation of organic precursors and the acidic chemical process. However, the temporal change of $R_{ra}$ did not follow a similar trend as the peak area of oxalic acid in most particles, possibly due to the multi-step formation of oxalic acid influenced by many factors such as precursors, liquid water content and ion strength (Carlton et al., 2007;Cheng et al., 2013;Cheng et al., 2015)." has been added in lines 428-455.

[Figure]

Figure 8. The temporal variations of peak area of nitrate, sulfate and oxalic acid, and the relative acidity ratio ($R_{ra}$) in carbonaceous type oxalic acid particles in winter.

The related discussions about Figure 9 and Figure S8 have been revised: "The organosulfate derived from glyoxal requires acidic aqueous environment of particles, and herein is used as an indicator of acid-catalyzed ageing process of organic compounds. The temporal variation of organosulfate (m/z=-155) containing particles during the entire sampling period in Heshan, China is shown in Figure S6. During the episode, oxalic acid particles had moderate linear correlation with organosulfate particles (Figure 9b)." has been changed to "The formation of organosulfates from glyoxal requires an acidic aqueous environment, which can be used as an indicator of acid-catalyzed ageing process of organic compounds. The temporal trend of organosulfate-containing oxalic acid particles in winter is shown in Figure S8, which exhibited a similar pattern as the total oxalic acid particles during the whole sampling period in winter. The percentage of organosulfate-containing oxalic acid particles in total oxalic acid particles ranged from 0 to 16.4% with the highest ratio observed in

the episode (February 8). The linear regression between total oxalic acid particles and organosulfate-containing oxalic acid particles in the episode is exhibited in Figure 9b, and the robust correlation ($r^2=0.81$) between them suggests that oxalic acid and organosulfate may share similar formation process." in lines 476-486.

[Figure]

Figure 9. The comprehensive study of oxalic acid particles increase on Feb 8, 2015: (a) The digitized positive and negative ion mass spectrum of oxalic acid particles during the episode; (b) Linear regression between total oxalic acid particles and organosulfate-containing oxalic acid particles (m/z -155).

[Figure]

Figure S8. Temporal variation of organosulfate (m/z=-155) containing particles in total particles and in oxalic acid particles in Heshan, China.

"The temporal trend of organosulfate-containing oxalic acid particles in winter is also shown in Figure S8, which exhibited a similar pattern as the total oxalic acid particles. The percentage of organosulfate-containing oxalic acid particles in total oxalic acid particles ranged from 0 to 16.4% with the highest ratio observed on February 8. " has been added in the supplement material.

**Minor comments**

1. L.83: Oxalic acid does not have a low vapor pressure. Its presence in ambient particles is due to salt and/or complex formation (cf also comment 1d).

**Response**: Oxalic acid is predominantly enriched in aerosols phase due to low vapor pressure ($3.5 \times 10^{-5}$ Torr) (Prenni et al., 2001) and high water solubility. Several field studies all indicated more than 70% of oxalic acid exist in aerosol phase (Limbeck et al., 2001;Mochida et al., 2003a;Mochida et al., 2003b). The salt and complex formation of oxalate with ammonium, potassium, sodium and metal ions have been considered as the main behavior of oxalic acid in the particle phase (Yao et al., 2002;Moffet et al., 2008;Furukawa and Takahashi, 2011), and a higher pH condition would be more favorable for the salt and complex formation. Due to the abundant signals of potassium, sodium and metal ions in this work, oxalic acid can react with these anions and stay in the particle phase. Besides, the oxalic acid particles and in-situ pH both exhibited an increase in the afternoon, which was in accordance with the announcement of more oxalate at higher pH.

2. L.119/120: There are several studies that have shown good agreement between predicted and measured oxalate levels [e.g., Wonaschuetz et al., 2012]

**Response**: This reference has been cited in the manuscript as "A number of ground based and airborne field studies have found a tight correlation between oxalic acid and sulfate in ambient particles and cloud droplets, relating aqueous phase chemistry to the formation of oxalic acid in aerosols and cloud droplets (Yao et al., 2002;Yao et al., 2003;Yu et al., 2005;Sorooshian et al., 2006;Sorooshian et al., 2007a;Sorooshian et al., 2007b;Miyazaki et al., 2009;Wonaschuetz et al., 2012;Wang et al., 2016)." in lines 99-104.

3. L. 121 and 122: These sentences are repetitive.

**Response**: "So far the formation mechanism of oxalic acid especially in urban areas is still not clear. Online measurements of the mixing state of oxalic acid provides a powerful context to better understand the formation of oxalic acid in aerosol particles and cloud droplets." has been changed to "While the formation mechanism of oxalic acid especially in urban areas is still not clear, online measurements of the mixing state of oxalic acid provide a powerful tool to better understand the formation of oxalic acid in aerosol particles and cloud droplets." in lines 122-125.

4. L. 240: The study by Sorooshian et al. focused mostly on the destruction of oxalalte in the presence of iron.

**Response**: "In summer HM type particles (purple) and total oxalic acid particles exhibited similar diurnal patterns, suggesting a possibly connection between the production of oxalic acid and the transition metals (e.g. Fe, Cu) (Sorooshian et al., 2013)." has been changed to "In summer HM type particles (orange color) and total

oxalic acid particles exhibited similar diurnal patterns, suggesting a possible connection between the production of oxalic acid and the transition metals (e.g. Fe, Cu) (Zhou et al., 2015)." in lines 269-271.

5. L. 264, and other places: 'Mixing ratio' usually refers to the ratio of molecules of one type to the total number of molecules (e.g. ppb = 1 in 10^9 molecules). The authors should change their wording as I assume here 'mixing ratio' is used in the meaning of 'number of particles that are internally mixed'.

**Response**: The "mixing ratio" has been replaced by "percentage" in the manuscript. "The mixing ratios of oxalic acid particles with sulfate, nitrate and ammonium (SNA) were investigated through the relative abundance of SNA-containing oxalic acid particles in total oxalic acid particles (Figure 3)." has been changed to "The mixing state of oxalic acid particles with sulfate, nitrate and ammonium (SNA) was investigated through the percentage of SNA-containing oxalic acid particles in total oxalic acid particles (Figure 3)." in lines 294-296.

"Oxalic acid was found to be internally mixed with sulfate and nitrate during both sampling periods with mixing ratio of 93% and 94% in summer respectively, and both 98% in winter (Figure 3 a)." has been changed to "Oxalic acid was found to be internally mixed with sulfate and nitrate during both sampling periods with percentage of 93% and 94% in summer respectively, and both 98% in winter (Figure 3 a)." in lines 296-298.

"However, the mixing ratio of $NH_4^+$ with oxalic acid was only 18% in summer but increased to 71% in winter. Linear correlations between $NH_4^+$-containing oxalic acid particles ($C_2$-$NH_4^+$) and total oxalic acid particles are depicted in Figure 3" has been revised to "However, the percentage of $NH_4^+$ with oxalic acid was only 18% in summer but increased to 71% in winter. Linear correlations between $NH_4^+$-containing oxalic acid particles ($C_2$-$NH_4^+$) and total oxalic acid particles are depicted in Figure 3" in lines 299-301.

"The low mixing ratio of $NH_4^+$ in oxalic acid particles in summer indicated the presence of oxalic acid in $NH_4^+$-poor particles." has been changed to "The low percentage of $NH_4^+$ in oxalic acid particles in summer indicated the presence of oxalic acid in $NH_4^+$-poor particles." in lines 302-304.

6. L. 289: Are all particle larger in winter than in summer or only those that contain oxalic acid?

**Response**: The discussion about size change only refers to oxalic acid containing particles.

7. L. 296-300: The mentioning of malonic acid is distracting here and does not lead to additional evidence or insights.

**Response**: Malonic acid is another product of photochemical oxidation of organic compounds according to many field studies (Kawamura and Ikushima, 1993;Wang et al., 2012;Meng et al., 2013;Meng et al., 2014), so we have discussed

the variation pattern of malonic acid in order to provide more evidence for the photochemical behavior of DCAs, thus we decide to keep this part.

8. L. 316: It is true that OH (like all other radicals) has a relatively short life time in the gas phase. However, the partitioning to the aqueous phase is limited due to its even shorter lifetime in the aqueous phase. Its solubility and the quick consumption in the aqueous phase leads to the limitation.

**Response**: "While the partition of •OH from gas to aqueous phase is limited by its low Henry's law constant ($K_{H,OH}$=30 M atm$^{-1}$) and short lifetime of •OH in the gas phase (Hanson et al., 1992), the main sources of aqueous •OH are from the photolysis of $H_2O_2$, $NO_3^-$, $NO_2^-$, and chromophoric dissolved organic matter (CDOM) (Yu et al., 2014;Badali et al., 2015;Gligorovski et al., 2015;Tong et al., 2016). Among these sources the photolysis of $H_2O_2$ through Fenton reactions involving the catalysis of transition metal ions like $Fe^{2+/3+}$, $Cu^{+/2+}$ and $Mn^{2+/3+}$ is an efficient source of •OH (Deguillaume et al., 2005;Herrmann et al., 2005;Ervens et al., 2014)." has been changed to "The main sources of •OH in the aqueous phase contain both direct uptake from the gas phase and the chemical sources in the aqueous phase such as Fenton type reactions and photolysis of $H_2O_2$, $NO_3^-$, $NO_2^-$, and chromophoric dissolved organic matter (CDOM) (Yu et al., 2014;Badali et al., 2015;Ervens, 2015;Gligorovski et al., 2015;Herrmann et al., 2015;Tong et al., 2016). Considering low Henry's law constant of •OH ($K_{H,OH}$=30 M atm$^{-1}$) (Hanson et al., 1992) and abundant fraction of transition metal ions in the oxalic acid particles, the photolysis of $H_2O_2$ through Fenton reactions involving the catalysis of transition metal ions like $Fe^{2+/3+}$, $Cu^{+/2+}$ and $Mn^{2+/3+}$ likely contributes substantially to the source of •OH in the aqueous phase in this work (Deguillaume et al., 2005;Herrmann et al., 2005;Ervens et al., 2014)." in lines 345-355.

9. L. 320: Fenton reaction is not a photolysis. In l. 337, it is stated correctly that Fenton reactions do not need necessarily light.

**Response**: We agree with the reviewer's comment. This point has been properly clarified in the manuscript.

10. L. 344: What are the influences of pH (is) from RH and inorganic ions? Figure S5 does not include any discussion.

**Response**: In the current work the $pH_{is}$ of ambient particles ranged from -1.42 to 4.01, and the influences of $pH_{is}$ from RH and inorganic ions are discussed in Figure S5. The related discussion has already been presented in the supplement as follows:

The $pH_{is}$ of ambient particles ranged from -1.42 to 4.01, which indicate that fine particles in the sampling site are highly acidic. These values are within the range of previous studies that investigated $pH_{is}$ through filter-based and real-time measurements in the PRD area (Pathak et al., 2004;Yao et al., 2006;Xue et al., 2011). Based on the calculation equation, the $pH_{is}$ is determined by the free amount of $H^+$ and liquid water content (LWC) in the aerosols. LWC is strongly dependent on the ambient RH and affected by water-soluble composition in the aerosols. Thus we

investigate the diurnal patterns of RH, major inorganic ions and the free amount of $H^+$ in Figure S5. Although RH increased from 00:00 to 07:00, $H^+$ had higher concentration during this period than the other time, which resulted the lower value of $pH_{is}$ (between -1.42 and 0) from 00:00 to 05:00. Thus more acidic aerosols with $pH_{is}$ below 0 were observed due to the combined effects of the RH and relative abundance of $H^+$ in aerosols.

11. L. 354: Not clear why 'on the other hand' as the following sentence is just another example of oxalate degradation.

**Response**: "On the other hand, photolysis of $Fe(oxalate)_n^{3-2n}$ can contribute to 99% of the overall degradation of oxalic acid" has been changed to "Furthermore, photolysis of $Fe(oxalate)_n^{3-2n}$ can contribute to 99% of the overall degradation of oxalic acid" in lines 387-388.

12. L. 439/440: This is a very strong and vague statement. How do the results of the current study help improving climate models and air pollution mitigation strategies?

**Response**: "The current study also indicates that SPAMS can be a robust tool for exploring the formation and transformation processes of SOA, contributing to the improvement of global climate modeling and the development of effective air pollution mitigation strategies." has been changed to "The current study demonstrates that SPAMS is a unique tool for understanding the mixing states of different components of ambient aerosols, which are useful for exploring the formation and evolution process of SOA." in lines 512-515.

Technical comments
13. L. 102: replace 'suggest' by 'suggested'
**Response**: The correction has been made in line 105.

14. L. 181: replace 'introducing' by 'introduced'
**Response**: The correction has been made in line 184.

15. L. 240: replace 'possibly' by 'possible'
**Response**: The correction has been made in line 270.

16. L. 273: remove 'that'
**Response**: The correction has been made in line 303.

[revised manuscript text omitted]

---

## Author Response (AR1)

**Response letter**

[Atmospheric Chemistry and Physics, MS ID: acp-2016-1081]
Title: Mixing state of oxalic acid containing particles in the rural area of Pearl River Delta, China: implication for seasonal formation mechanism of Secondary Organic Aerosol (SOA)

Dear editor

Thank you for your comments and contribution to our manuscript. We have responded all the comments from three anonymous referees, and have submitted the point-by-point response to the interactive discussion. We appreciate these valuable and helpful comments to our work and enable it to meet the high quality of the journal Atmos. Chem. Phys. In addition to the relevant changes to all the comments from three anonymous referees, several modifications have also been made in the manuscript and listed below. The marked-up manuscript has been uploaded.

Anything about our paper, please feel free to contact me at limei2007@163.com

Best regards!

Sincerely yours

Mei Li
April 7, 2017

**Several modifications in the manuscript:**

1. We have added two co-authors in the manuscript due to their helpful discussions. The author list "Chunlei Cheng[1,2], Mei Li[1,2*], Chak K. Chan[3], Haijie Tong[4], Changhong Chen[5], Duohong Chen[6], Dui Wu[1,2], Lei Li[1,2], Peng Cheng[1,2], Wei Gao[1,2], Zhengxu Huang[1,2], Xue Li[1,2], Zhong Fu[7], Yanru Bi[7], Zhen Zhou[1,2*]" has been changed to "Chunlei Cheng[1,2], Mei Li[1,2*], Chak K. Chan[3], Haijie Tong[4], Changhong Chen[5], Duohong Chen[6], Dui Wu[1,2], Lei Li[1,2], Cheng Wu[1,2], Peng Cheng[1,2], Wei Gao[1,2], Zhengxu Huang[1,2], Xue Li[1,2], Zhijuan Zhang[1,2], Zhong Fu[7], Yanru Bi[7], Zhen Zhou[1,2*]".

2. We have changed the keywords "Secondary organic aerosols" to "Aqueous phase reactions" in line 69.

3. "Haijie Tong acknowledge Max Planck Society for funding and Ulrich Pöschl for helpful discussions." has been added in the acknowledgements in lines 525-526.

---

## Referee Report (RR1)

Re-Review by the manuscript by Cheng et al.,
The authors have clarified a few of my previous comments. However, several of my original concerns were not addressed at all or only very poorly.
I list my main concerns below and also comment on the authors' response to my previous comments in order to highlight where more detail is needed.

**Major comments**

**1) Role of Fenton chemistry**
I still find the discussion of the role of iron chemistry in SOA formation confusing. It is possible that Fenton chemistry enhances OH concentration. However, there are many other radical sources that are not discussed.

**2) Role of metal in decreasing SOA**
Neither in the abstract nor in the conclusions, have the authors discussed the significant role of Fe-oxalato complexes as a sink of oxalate. Instead, the only modified slightly a couple sentences in the main parts of the manuscript without discussing in detail the balance between oxalate formation and loss.

**3) Acid catalysis**
By definition, a catalysis is a process where a compound (a catalyst) is involved in the reaction but will be recycled and not be consumed. The authors refer to 'acid-catalysed oxalate formation from glyoxal' (l. 447). These studies did not show acid catalyzed reactions.

**4) Trivialities**
At several places, the authors include sentences that do not add to the discussion but are circular and trivial. Examples include
 - l. 302: The low percentage of $NH_4^+$ in oxalic acid particles in summer indicated the presence of oxalic acid in $NH_4^+$ poor particles.
… do you want to say here that oxalic acid is rather accumulated in particles that do not contain $NH_4^+$?

 - l. 438: Nitrate, sulfate and oxalic acid showed very similar variation patterns in winter, indicating a close connection of their coexistence.
… what is meant by 'connection of coexistence'? – it seems redundant.

**Minor comments**

l. 60: 'relative acidity ratio' should be deifned here (or simply called 'acidity')

l. 111-112: The mechanism of oxalic acid acid as described in the manuscript is not uncertain. The main uncertainty are the oxidant levels - as correctly stated in the response to the reviews, but is should be added here.

**l. 369-371: This is completely vague and does not have any observational basis. This should be accompanied by a more detailed analysis.**

**l. 378: The figure merely shows RH and some inorganic compounds etc. The figure does not 'discuss' anything but only shows time traces. In addition, I suggest showing the H+ concentration in this figures as mol/L(aq) so that it can help the discussion of possible acidity effects.**

**l. 394: replace 'due to reported studies' by 'according to previous studies' (or something similar)**

**l. 414: During haze periods, photochemical activity is usually reduced.**

**l. 429: Remove 'particles' here. 'Oxalic acid particles are prevalent in carbonaceous particles' sounds awkward.**
* * *
**Below is the response by the authors to my original comment. My added comments are in red.**

**Specific comments and point by point responses:**

Major comments

1. The suggested chemical mechanism is oversimplified and contains several misconceptions and/or omissions:

a) Generally, it is assumed that direct OH uptake is the main source of OH in the aqueous phase [e.g., Ervens et al., 2003; Herrmann, 2003; Tilgner et al., 2013]. Therefore higher OH (gas) concentration will lead to higher OH concentration in the aqueous phase. Higher OH (aq) concentration will also lead to more oxidation of oxalate and therefore less SOA.

**Response**: We thank the reviewer‟s comments. The main sources of •OH in the aqueous phase contain both direct uptake from the gas phase and chemical sources in the aqueous phase such as Fenton reactions and photolysis of $H_2O_2$ and nitrate (Ervens et al., 2014;Ervens, 2015;Gligorovski et al., 2015;Herrmann et al., 2015). Although the concentrations of $H_2O_2$ and •OH in gas and aqueous phases are not available in this work, considering much higher Henry‟s law constant of $H_2O_2$ ($K_H=8.3\times10^4$ M atm$^{-1}$) than •OH ($K_H=30$ M atm$^{-1}$) (Hanson et al., 1992;O'Sullivan et al., 1996) and abundant fraction of transition metals in the oxalic acid particles in this work, aqueous phase chemistry likely contributes substantially to the source of •OH in the aqueous phase according to high efficiency of •OH production from Fenton type reactions (Gligorovski et al., 2015). The aqueous phase oxidation of glyoxal and methylglyoxal is the main source of oxalic acid production, and their reaction rates with •OH ($1.1\times10^9$ mole$^{-1}$ s$^{-1}$) is one order of magnitude higher than oxalic acid with •OH ($1.9\times10^8$ mole$^{-1}$ s$^{-1}$) as previously reported (Herrmann, 2003;Myriokefalitakis et al., 2011). Thus, the high •OH (aq) concentration would lead to the enrichment of oxalic acid in aqueous phase.

In order to clarify these points, we have changed "While the partition of •OH from gas to aqueous phase is limited by its low Henry"s law constant ($K_{H,OH}$=30 M atm$^{-1}$) and short lifetime of •OH in the gas phase (Hanson et al., 1992), the main sources of aqueous •OH are from the photolysis of $H_2O_2$, $NO_3^-$, $NO_2^-$, and chromophoric dissolved organic matter (CDOM) (Yu et al., 2014;Badali et al., 2015;Gligorovski et al., 2015;Tong et al., 2016). Among these sources the photolysis of $H_2O_2$ through Fenton reactions involving the catalysis of transition metal ions like $Fe^{2+/3+}$, $Cu^{+/2+}$ and $Mn^{2+/3+}$ is an efficient source of •OH (Deguillaume et al., 2005;Herrmann et al., 2005;Ervens et al., 2014)." to "The main sources of •OH in the aqueous phase contain both direct uptake from the gas phase and the chemical sources in the aqueous phase such as Fenton type reactions and photolysis of $H_2O_2$, $NO_3^-$, $NO_2^-$, and chromophoric dissolved organic matter (CDOM) (Yu et al., 2014;Badali et al., 2015;Ervens, 2015;Gligorovski et al., 2015;Herrmann et al., 2015;Tong et al., 2016). Considering the low Henry"s law constant of •OH ($K_{H,OH}$=30 M atm$^{-1}$) (Hanson et al., 1992) and abundant fraction of transition metal ions in the oxalic acid particles, the photolysis of $H_2O_2$ through Fenton reactions involving the catalysis of transition metal ions like $Fe^{2+/3+}$, $Cu^{+/2+}$ and $Mn^{2+/3+}$ likely contributes substantially to the source of •OH in the aqueous phase in this work (Deguillaume et al., 2005;Herrmann et al., 2005;Ervens et al., 2014)." in lines 345-355.

Reviewer, 2nd round: The Fenton reaction is NOT a photolysis of H2O2. A photolysis would yield two OH radicals, whereas Fenton reaction leads to OH + OH-

b) Higher iron concentration might lead to more OH. However, more importantly is the effect of the loss of oxalate due to the photolysis of the iron-oxalato complex. While this reaction is mentioned in the manuscript, its predominating role in oxalate loss [Sorooshian et al., 2013] is not discussed in a balanced way.

**Response**: The degradation of oxalic acid from the complex with iron has been discussed in the manuscript to explain the diurnal change of oxalic acid particles. Indeed, it would be more specific to evaluate the net production of oxalic acid from photochemical process if we could calculate the production and loss of oxalic acid. Unfortunately, we have no real time data about the ambient concentrations of oxalic acid and iron, so the exact amount of oxalic acid loss from the photolysis of the iron complexes cannot be obtained for now.

Reviewer, 2nd round: l. 383-392: While the authors add some text about the significant loss of oxalic acid due to the photolysis of the iron-oxalato-complex, this discussion is neither balanced nor well connected with the rest of the discussion. If iron is indeed abundantly available for Fenton reaction, why doesn't it show an impact on oxalic acid during day time when - according to literature - up to 99% of oxalic acid might be reduced?

c) At low pH, it can be expected that reaction rates are lower since in general the undissociated acids (glyoxylic, glycolic) react more slowly than their dissociated

counterparts. Oxalate has a very low pKa (1.23) so that even at low pH a substantial fraction is still present as oxalate. Could changes in pH and therefore reaction rates explain some of the temporal trends?

**Response**: Indeed, the reaction rate of glyoxylate with •OH ($k=2.8\times10^9$ mole$^{-1}$ s$^{-1}$, pH=8) is one order of magnitude higher than glyoxylic acid ($k=3.6\times10^8$ mole$^{-1}$ s$^{-1}$, pH=1), so the aqueous phase with high pH value is more favorable for the production of oxalic acid from the oxidation of precursors. The SPAMS can only detect the ion peak of oxalate ion (m/z -89) in single particles, so the percentage of oxalate in total oxalic acid cannot tell from the data of SPAMS. We believe the diurnal change of pH also has influence on the oxidative process of organic precursors in addition to provide a proper environment for Fenton like reactions. We have added the related discussion in the manuscript. "In addition to the contribution from Fenton reactions after 12:00, the precursors of oxalic acid such as glyoxylic acid have higher reaction rate with •OH at high pH based on previous studies(Ervens et al., 2003;Herrmann, 2003;Cheng et al., 2015), thus the increase of pH not only enhances •OH production from photo-Fenton reactions, but also promotes the oxidation process of the precursors of oxalic acid by •OH." has been added in lines 392-397.

Reviewer, 2nd round: The same is true for the loss of oxalic acid by OH reactions. This should be mentioned here.

d) At very low pH (< 1.23), it is expected that oxalic acid is present in undissociated form and therefore not able to make salts or complexes that „trap" it in the particle phase. This fact contradicts the trend of increased oxalate concentrations at low pH. This should be discussed.

**Response**: Oxalic acid is predominantly enriched in particle phase due to its low vapor pressure ($3.5\times10^{-5}$ Torr) (Prenni et al., 2001) and high water solubility. Several studies have indicated that more than 70% of oxalic acid could exist in particle phase (Limbeck et al., 2001;Mochida et al., 2003a;Mochida et al., 2003b). The salt and complex formation of oxalate with ammonium, potassium, sodium and metal ions have been considered as the main behavior of oxalic acid in the particle phase (Yao et al., 2002;Moffet et al., 2008;Furukawa and Takahashi, 2011), and a higher pH condition would be more favorable for the salt and complex formation. Due to the abundant signals of potassium, sodium and metal ions in this work, oxalic acid can react with these anions and stay in the particle phase. Besides, the oxalic acid particles and in-situ pH both exhibited an increase in the afternoon, which was in accordance with the announcement of more oxalate at higher pH.

Reviewer, 2nd round: At low pH, when oxalic acid is not dissociated, the Henry's law constant is 9000 M/atm. That means only a very small fraction (<<1%) of all oxalic acid is expected to be in the aqueous phase of particles. Therefore, the authors' arguments here are not convincing.

e) The proposed mechanism is by no means new or detailed (l. 360). It does not include any sinks of oxalate, nor complex formation. It is one possible formation mechanism of oxalic acid from glyoxal. The generalization to dicarbonyls and aldehydes is not correct since only small compounds (C2) will follow the suggested reaction pathways.

**Response**: The photochemical production of oxalic acid has been studied in many field researches due to a similar diurnal pattern with $O_3$ (Kawamura and Ikushima, 1993;Kawamura and Yasui, 2005;Aggarwal and Kawamura, 2008;Miyazaki et al., 2009), but the detailed photochemical formation process of oxalic acid has not been comprehensively discussed. While the oxidation of glyoxal and glyoxylic acid has been proposed and discussed in many field, laboratory and model studies (Myriokefalitakis et al., 2011;Wang et al., 2012;Kawamura et al., 2013;Ervens et al., 2014;Wang et al., 2014), the influential factors of their oxidation reactions with •OH still needs to be ascertained from the field studies. Based on a comprehensive discussion of the mixing state of oxalic acid with secondary ions and transition metals, we propose a plausible explanation to connect among the diurnal pattern of $O_3$, oxalic acid, iron and in-situ pH. (Sorooshian et al., 2013;Zhou et al., 2015). However, without the direct measurement of the concentrations of •OH, iron ions, and oxalic acid, the proposed photochemical formation mechanism of oxalic acid in the original manuscript is difficult to be confirmed. Thus, we have changed the expression about the oxalic acid production in HM type particles into a plausible explanation instead of a proposed formation mechanism, and have softened our tone in the discussions. The schematic diagram to explain the formation process of oxalic acid in the HM type particles has been moved to the supplement material in Figure S6.

The related discussion has been revised as follows:

In the abstract "Furthermore, favorable in-situ pH (2-4) conditions were observed, which promote Fenton like reactions for efficient production of •OH in HM type particles. A mechanism in which products of photochemical oxidation of VOCs partitioned into the aqueous phase of HM particles, followed by multistep oxidation of •OH through Fenton like reactions to form oxalic acid is proposed." has been changed to "The favorable in-situ pH (2-4) and the dominance of transition metal ions in oxalic acid particles can be plausibly explained by the enhanced production of •OH from Fenton like reaction, which can promote the oxalic acid production from the oxidation of precursors by •OH in HM type particles." in lines 55-58.

"Based on above discussions, detailed mechanism for oxalic acid formation in acidic aqueous phase of particles is proposed for our field observations (Figure 8)." has been changed to "Based on above discussions of the mixing state of oxalic acid with secondary ions and transition metals, a plausible explanation to the formation process of oxalic acid in the HM type oxalic acid particles is proposed (Figure S6)." in lines 398-400.

In the conclusion "The favorable in-situ pH and the dominance of transition metal ions in oxalic acid particles suggests an enhanced production of •OH from Fenton like reactions. A mechanism involving the photochemical production of VOCs via efficient aqueous phase reactions with enhanced •OH to oxalic acid was proposed." has been

changed to "Furthermore, suitable in-situ pH is favorable for Fenton like reactions to produce •OH in HM type particles, and might promote the oxalic acid production from the oxidation of precursors by •OH in HM type particles." in lines 504-506.

To address the referee''s concern, we have revised the description of dicarbonyls and aldehydes in Figure S6. The "dicarbonyls and aldehydes" has been changed to "Low molecular weight dicarbonyls (e.g. glyoxal)", and "Hydrated dicarbonyls and aldehydes" has been changed to "Hydrated dicarbonyls (e.g. glyoxal)" in Figure S6:

[Figure]

Figure S6. A schematic diagram to explain the formation process of oxalic acid in the HM type particles: the red steps are enhanced by photochemical activities in the current study.

Reviewer, 2nd round: In this figure, the loss of oxalate due to the photolysis of the iron-oxalato complex is missing.

2. The number fraction of oxalate containing particles seems very low. Is this comparable to other measurements? What was the mass fraction of oxalate (a) in the particles and (b) related to the total aerosol loading?

**Response**: Yang et al. (2009) has measured the oxalic acid-containing particles in the urban area of Shanghai by ATOFMS and found 15,789 oxalate-containing particles, accounting for 3.4% of the total collected particles. In this work 13,109 and 20,504 of oxalic acid-containing particles were obtained in summer and winter separately, accounting for 2.5% and 2.7% of the total detected particles. The abundance of oxalic acid-containing particles in this work was lower than the reported studies in the urban area of Shanghai (3.4%), which was possibly due to less anthropogenic precursors for oxalic acid at the rural sampling site in Heshan. Higher abundance of oxalic acid particles (1-40%) was observed in the much cleaner western Pacific Ocean by Sullivan et al. (2007), which corresponded to higher ambient concentration of DCAs

(19±4.8%) in total particulate organic matter. From the reported studies in PRD (Yao et al. 2004; Ho et al. 2011), the abundance of DCAs was 1-3.5% in total organic matter, which was much lower than those in the western Pacific Ocean, leading to lower percentage of oxalic acid-containing particles in this work. We have added the comparison between this work and the study in Shanghai in the manuscript. "The percentage of oxalic acid-containing particles in total particles in this work was comparable to the reported value in the urban area of Shanghai (3.4%) (Yang et al., 2009). However, these percentages are in general much lower than those reported in cleaner environments such as the western Pacific Ocean where oxalic acid was found in up to 1-40% of total particles due to little anthropogenic influences (Sullivan and Prather, 2007)." is added in lines 192-198.

For the second question, the relative fraction of oxalic acid particles in total detected particles was accounting for 2.5% and 2.7% of the total detected particles in summer and winter (Figure 1a). However, because we didn"t collect filter samples during the sampling period, we cannot calculate the relative mass fraction of oxalic acid in ambient $PM_{2.5}$.

Reviewer, 2nd round: Thanks for clarifying.

3. Was all iron in the particles in form of soluble iron, i.e. available for reaction?

**Response**: Indeed, it would be more convincing to discuss the role of Fenton reactions in the formation of oxalic acid particles if we could obtain the concentration of $Fe^{2+/3+}$ ions. However, we didn"t collect filter samples during the sampling period, so the concentration of $Fe^{2+/3+}$ ions were not measured. Besides, the SPAMS could only detect the peak area of iron element, and the different valence state of iron could not be distinguished. Therefore, the discussions between Fenton reactions and oxalic acid production have been revised to a plausible explanation instead of a proposed formation mechanism.

Reviewer, 2nd round: Given all these uncertainties and the fact that the authors do not have any measurements on iron availability, I suggest to word the role of Fenton chemistry much more carefully throughout the whole manuscript.

4. Oxalate and the other DCAs usually represent only a very small fraction of the total organic aerosol mass. Therefore the title is misleading as it talks about SOA in general.

**Response**: Based on the comments from you and other reviews, the title "Mixing state of oxalic acid containing particles in the rural area of Pearl River Delta, China: implication for seasonal formation mechanism of Secondary Organic Aerosol (SOA)" has been changed to "Mixing state of oxalic acid containing particles in the rural area of Pearl River Delta, China: implications for the formation mechanism of oxalic acid".

Reviewer, 2nd round: I appreciate this change.

5. It seems based on Figure S5, that RH was always < 100% (except a very brief period). Therefore, the discussed aqueous chemistry will have to take place in aqueous

aerosol. There are many studies that have discussed different reaction pathways in aqueous aerosol vs cloud [e.g., Tan et al., 2009; Lim et al., 2010] with less efficient oxalate formation in the former. In addition, it seems likely that iron ions might be less dissolved in the rather highly concentrated aqueous aerosol solutions. All discussion is about chemistry as it happens in cloud droplets. These two regimes should be differentiated.

**Response**: While the aqueous phase formation of oxalic acid from organic precursors has been found both in cloud droplets and aerosols water, the more efficient production in cloud droplets than aerosols water has been reported by several studies (Tan et al., 2009;Lim et al., 2010;Myriokefalitakis et al., 2011). In this work, the well internally mixing state between oxalic acid and sulfate in summer and winter suggests a common production route of oxalic acid and sulfate, likely aqueous phase reactions. The strong photochemical pattern of oxalic acid particles with $O_3$ under the condition of RH below 100% implies a less influence from cloud-processing of oxalic acid particles. Thus we speculate the oxalic acid formation process in HM type particles is related to aqueous phase reactions in aerosols, and this point has been stated in the manuscript: "Based on above discussions of the mixing state of oxalic acid with secondary ions and transition metals, a plausible explanation to the formation process of oxalic acid in the HM type oxalic acid particles is proposed (Figure S6). In summer strong photochemical activity and high $O_3$ concentrations in the afternoon lead to more production of reactive radicals such as •OH and $HO_2^•$, which promote the oxidation of VOCs to dicarbonyls and aldehydes (e.g. glyoxal and methylglyoxal), followed by a subsequent partitioning into the aqueous phase of particles" in lines 398-404.

Although the reviewer has pointed out that iron ions might be less dissolved in the rather highly concentrated aqueous aerosol solutions, the low in-situ pH of aerosols is also favorable for the dissolved of iron. Because we didn"t collect filter samples during the sampling period, the concentration of $Fe^{2+/3+}$ ions were not measured, so the ratio of $Fe^{2+/3+}$ ions to total iron element could not be evaluated for now, and we will further discuss this issue in our next study.

Reviewer, 2nd round: The studies by Tan et al., Lim et al. and other by the Turpin group and others, have shown that oxalic acid is not efficiently formed in aqueous aerosol particles. – as I had pointed to in my previous comment.
Either the authors need to clarify that despite of RH < 100% (i.e. no clouds) at the study location, the observed oxalic acid was likely formed in clouds prior to arriving at the study location (what data is available to prove cloudiness) or they should at least briefly mention the more significant formation of oligomers as shown in the cited references.

6. While briefly discussed, it is not clear to what extent different air masses cause different oxalate levels. How much of the measured oxalate is background material? Did other meteorological conditions affect the concentrations such as changes in boundary layer?

**Response**: The different air masses arriving in sampling site in summer and winter had substantial impact on the amount of organic precursors from anthropogenic emissions. More carbonaceous signals were found in the oxalic acid particles in winter since oxalic acid is mainly derived from secondary oxidation of VOCs and subsequent intermediates. However, it is still difficult to quantify exactly the increase of oxalic acid concentration in winter based on the SPAMS data. The sampling site in this work is a rural site and the aerosols are influenced by the regional transport in PRD. Although we cannot obtain the exact background level of oxalic acid, the lowest percentage of oxalic acid particles in total detected particles is used to evaluate the approximate background level of oxalic acid particles. The lowest hourly percentage of oxalic acid particles in total detected particles were 0.1% and 0.5 % in summer and winter, which were much lower than the average value in summer (2.3%) and winter (2.8%), suggesting a small impact of the background distribution to the measurements of oxalic acid particles.

The wind speed was from 0.3 to 4.0 m·s$^{-1}$ with an average of 1.6 m·s$^{-1}$ during the sampling period in winter, which indicated a rather stagnant atmospheric condition. The stagnant atmospheric condition in winter was favorable for the aging process of aerosols and might have contributed to the broader size distribution of oxalic acid particles. Unfortunately the height of boundary layer was not available in this work. Nevertheless, the relative abundance of oxalic acid particles had no obvious increase at night and the lower mixing height may not be an important factor during the formation process of oxalic acid in the winter. We will further examine their relationships in our next study.

The diurnal variations of meteorological factors such as temperature, RH and wind speed have been discussed in Figure S7.

[Figure]

Figure S7. The diurnal variations of temperature (T), RH, wind speed (WS), oxalic acid particles, total EC particles, the EC type oxalic acid-containing particles and ambient NO$_2$ concentrations from July 28 to August 1 in 2014.

We have added the related discussion in the manuscript. "The diurnal patterns of temperature, wind speed are presented in Figure S7. The high temperature between

9:00 and 19:00 was favorable to the secondary processing of organic precursors. The wind speed was low during the whole day, especially between 9:00 and 18:00, which provided a stagnant environment for the increase in oxalic acid produced from photochemical process. The influence from traffic emission was investigated through the diurnal variations of total EC type particles and $NO_2$ (Figure S7). The EC type particles increased from 12:00 to 21:00, which had same variation as total oxalic acid, but $NO_2$ followed the rush hour pattern with two peaks from 5:00 to 8:00 and from 18:00 to 21:00. Traffic emission is not expected to have a large contribution to oxalic acid in this study." has been added in lines 417-426 in the manuscript.

Reviewer, 2nd round: Why is higher temperature favorable for more SOA formation? Rate constants usually increase. However, they do so for both formation rates and loss rates of oxalic acid. In addition, higher temperature also favor higher evaporation of volatile gases, including water.

7. I am not sure what Figure S6 is really showing. Does it show a correlation of organosulfur particles and oxalate or does it simply show that more particles cause higher concentrations of „everything"? How about the mixing state of organosulfur compounds and oxalic acid particles? The fact that they are in the same particle class, does not necessarily mean that they are internally mixed and therefore their formation pathways are related.

**Response**: The original Figure S6 (now Figure S8) presented the temporal variation of total organosulfate (m/z=-155) containing particles during whole sampling periods in Heshan, China. The temporal trend of organosulfate-containing oxalic acid particles in winter has been added in Figure S8, which exhibited a similar pattern as the total oxalic acid particles. The percentage of organosulfate-containing oxalic acid particles in total oxalic acid particles ranged from 0 to 16.4% with the highest ratio observed on February 8. The linear regression between oxalic acid particles and organosulfate particles in Figure 9b has been replaced by the correlation between organosulfate-containing oxalic acid particles and total oxalic acid particles, and the robust correlation ($r^2$=0.81) between them supports a possible production of oxalic acid from acid-catalyzed reactions.

We also believe more evidence and discussion are needed to support the connection between oxalic acid formation process and acidic aqueous phase chemistry. So the related discussions have added as follows:

"Despite lower $O_3$ concentrations and photochemical activity in winter, oxalic acid particles were still prevalent in carbonaceous particles, especially BB type particles. While oxalic acid was found to be internally mixed with sulfate and nitrate both in summer and winter, the nitric acid was only observed in oxalic acid particles in winter, indicating a strongly acidic nature of oxalic acid particles in winter. Considering a possible connection of oxalic acid production with the acidic environment, the temporal concentrations of oxalic acid, sulfate and nitrate were investigated through their peak areas in the carbonaceous type oxalic acid particles including EC, OC, ECOC and BB

type in Figure 8. The peaks of m/z -62[$NO_3$]$^-$ and - 97[$HSO_4$]$^-$ represent nitrate and sulfate, respectively. Nitrate, sulfate and oxalic acid showed very similar variation patterns in winter, indicating a close connection of their co-existence. Although nitric acid was found in the oxalic acid particles, the acidity of the oxalic acid particles was not estimated since the real-time concentration of inorganic ions was not available during the sampling period in winter. Instead the relative acidity ratio ($R_{ra}$), defined as the ratio of total peak areas of nitrate and sulfate to the peak area of ammonium (m/z 18[$NH_4$]$^+$), was used (Denkenberger et al., 2007;Pratt et al., 2009). The $R_{ra}$ of carbonaceous type oxalic acid particles ranged from 7 to 114 with an average value of 25 (Figure 8), indicating an intensely acidic environment of carbonaceous type oxalic acid particles in winter. Several studies have reported potential production of oxalic acid from acid-catalyzed aqueous phase reactions in aerosols (Carlton et al., 2006;Carlton et al., 2007;Tan et al., 2009). In this work the acidic environment of the carbonaceous type oxalic acid particles and similar variation patterns among oxalic acid, sulfate and nitrate may suggest a relationship between the degradation of organic precursors and the acidic chemical process. However, the temporal change of $R_{ra}$ did not follow a similar trend as the peak area of oxalic acid in most particles, possibly due to the multi-step formation of oxalic acid influenced by many factors such as precursors, liquid water content and ion strength (Carlton et al., 2007;Cheng et al., 2013;Cheng et al., 2015)." has been added in lines 428-455.

[Figure]

Figure 8. The temporal variations of peak area of nitrate, sulfate and oxalic acid, and the relative acidity ratio ($R_{ra}$) in carbonaceous type oxalic acid particles in winter.

The related discussions about Figure 9 and Figure S8 have been revised: "The organosulfate derived from glyoxal requires acidic aqueous environment of particles, and herein is used as an indicator of acid-catalyzed ageing process of organic compounds. The temporal variation of organosulfate (m/z=-155) containing particles during the entire sampling period in Heshan, China is shown in Figure S6. During the episode, oxalic acid particles had moderate linear correlation with organosulfate particles (Figure 9b)." has been changed to "The formation of organosulfates from glyoxal requires an acidic aqueous environment, which can be used as an indicator of acid-catalyzed ageing process of organic compounds. The temporal trend of

organosulfate-containing oxalic acid particles in winter is shown in Figure S8, which exhibited a similar pattern as the total oxalic acid particles during the whole sampling period in winter. The percentage of organosulfate-containing oxalic acid particles in total oxalic acid particles ranged from 0 to 16.4% with the highest ratio observed in the episode (February 8). The linear regression between total oxalic acid particles and organosulfate-containing oxalic acid particles in the episode is exhibited in Figure 9b, and the robust correlation ($r^2$=0.81) between them suggests that oxalic acid and organosulfate may share similar formation process." in lines 476-486.

[Figure]

Figure 9. The comprehensive study of oxalic acid particles increase on Feb 8, 2015: (a) The digitized positive and negative ion mass spectrum of oxalic acid particles during the episode; (b) Linear regression between total oxalic acid particles and organosulfate-containing oxalic acid particles (m/z -155).

[Figure]

Figure S8. Temporal variation of organosulfate (m/z=-155) containing particles in total particles and in oxalic acid particles in Heshan, China.

"The temporal trend of organosulfate-containing oxalic acid particles in winter is also shown in Figure S8, which exhibited a similar pattern as the total oxalic acid particles. The percentage of organosulfate-containing oxalic acid particles in total oxalic

acid particles ranged from 0 to 16.4% with the highest ratio observed on February 8. ” has been added in the supplement material.

Reviewer, 2nd round: Thanks for clarification.

---

## Author Response (AR2)

**Response to the comments**

[Atmospheric Chemistry and Physics, MS ID: acp-2016-1081]
Title: Mixing state of oxalic acid containing particles in the rural area of Pearl River Delta, China: implications for the formation mechanism of oxalic acid

**General comments:**
The authors have clarified a few of my previous comments. However, several of my original concerns were not addressed at all or only very poorly. I list my main concerns below and also comment on the authors' response to my previous comments in order to highlight where more detail is needed.

**Response**: Many thanks for your $1^{st}$ and $2^{nd}$ round reviewing and comments. Indeed, it will be great to give a comprehensive and quantitative discussion on the contributions of various source •OH radicals especially that formed through Fenton reactions to the formation of liquid phase oxalic acid. However, the SPAMS could only detect the peak area of iron element, and the concentration of $Fe^{2+/3+}$ ions as well as $H_2O_2$ could not be obtained in the current work. Here, we prefer to cut down the Fenton reaction discussion to minimum as a possibility according to the literature, and the proposed mechanism has also been completely removed from the whole manuscript along the advice of the referee. Furthermore, we have also extended the discussion of the potential sink of oxalic acid via the photolysis of iron oxalato complexes corresponding to the diurnal variation of peak area of iron and oxalic acid in HM type particles. Finally we appreciate your comments and response that enable current study to meet the high quality of the journal *Atmos. Chem. Phys*., and our point by point response to your comments as well as our revisions have been shown in the following section.

Anything about our paper, please feel free to contact me at limei2007@163.com

Best regards!

Sincerely yours
Mei Li
May 24, 2017

**Specific comments and point by point responses:**

**Major comments**

1) Role of Fenton chemistry

I still find the discussion of the role of iron chemistry in SOA formation confusing. It is possible that Fenton chemistry enhances OH concentration. However, there are many other radical sources that are not discussed.

**Response**: According to the referee's 1[st] and 2[nd] round comments, we think that more field measurements and data are needed to comprehensively and qualitatively discuss the contribution of Fenton reaction source •OH radicals to oxalic acid formation. This is out of the interest of this work. Thus we prefer to remove the discussion of the contribution of Fenton reaction in the formation process of oxalic acid and the proposed mechanism along the suggestion of the referee. Fenton reaction is only mentioned as a possibility according to the literature in a very cautious expression.

[revised manuscript text omitted]
. While the aqueous phase oxidation of glyoxal can both take place in clouds and wet aerosols (Lim et al., 2010), the lower yield of oxalic acid from glyoxal in wet aerosols compared to clouds has been reported in chamber experiments due to the significant formation of higher molecular weight products such as oligomers in aerosols-relevant concentration (Carlton et al., 2007;Tan et al., 2009), which helps to explain the lower peak of oxalic acid particles at 15:00 compared to 19:00. Besides, the precursors of oxalic acid such as glyoxylic acid have higher reaction rate with •OH in high pH solutions according to previous studies(Ervens et al., 2003;Herrmann, 2003;Cheng et al., 2015), and in this work the increase of $pH_{is}$ was observed as the enhancement of oxalic acid particles in the afternoon (Figure 7), which suggests an efficient oxalic acid production from the

oxidation of precursors.

The similar photochemical pattern of HM type particles with $O_3$ and total oxalic acid particles implies a possible participation of metal ions in the formation process of oxalic acid. The modeling studies from Ervens et al. (2014) suggest that oxalic acid production from glyoxal and glyoxylic acid in aqueous phase significantly depends on •OH availability (Ervens et al., 2014). The main sources of aqueous phase •OH in cloud droplets include direct uptake from the gas phase(Jacob, 1986), ozone photolysis by UV and visible light at the air-water interface (Anglada et al., 2014), and also aqueous phase chemical reactions(Gligorovski et al., 2015). For the last kind of source, •OH radicals could be generated through Fenton or Fenton like reactions and photolysis of $H_2O_2$, $NO_3^-$, $NO_2^-$, and chromophoric dissolved organic matter (CDOM) (Badali et al., 2015;Ervens, 2015;Herrmann et al., 2015;Tong et al., 2016). Given that SPAMS cannot be used to quantify the concentrations of iron ions and $H_2O_2$, we will investigate the relative contribution of different source •OH radicals to the formation of oxalic acid and show results in our follow up studies." in lines 335-373.

2) Role of metal in decreasing SOA

Neither in the abstract nor in the conclusions, have the authors discussed the significant role of Fe-oxalato complexes as a sink of oxalate. Instead, the only modified slightly a couple sentences in the main parts of the manuscript without discussing in detail the balance between oxalate formation and loss.

**Response**: The sink of oxalic acid via the photolysis of iron oxalato complexes has been discussed through the diurnal variation of peak area of iron and oxalic acid in HM type particles.

"The number concentration of oxalic acid particles peaked at 19:00 instead of during the strong photochemical activity period in the afternoon; this was possibly due to the efficient degradation of oxalic acid from the complex with iron (Sorooshian et al., 2013;Zhou et al., 2015). Furthermore, photolysis of $Fe(oxalate)_n^{3-2n}$ can contribute to 99% of the overall degradation of oxalic acid (Weller et al., 2014). Although the enhanced •OH production from photo-Fenton reactions was favorable for the formation of oxalic acid from 12:00 to 18:00, we speculate that a high degradation rate of oxalic acid by iron complexation resulted in a lower net production of oxalic acid than at 19:00." has been revised to "The oxalic acid loss through the photolysis of iron oxalato complexes is a significant sink according to field measurements and model simulations (Sorooshian et al., 2013;Weller et al., 2014;Zhou et al., 2015). Considering the high abundance of iron in oxalic acid particles in the current work (Figure 6), the photolysis of iron oxalato complexes could have played important role in the diurnal variation of oxalic acid particles. Because the mass concentration of Fe (III) and oxalic acid could not be obtained through SPAMS, the diurnal variations of peak area of iron (m/z=56) and oxalic acid (m/z=-89) were used to investigate the role of iron on the net production of oxalic in

the HM type particles from July 28 to August 1, 2014 (Figure 8). Interestingly, the peak area of iron likely anti-correlated with the peak area of oxalic acid from 4:00 to 11:00. As the peak area of Fe increased from 1565 to 29920 from 4:00 to 7:00, the peak area of oxalic decreased from 6052 to 3487 accordingly. From 8:00 to 11:00, the peak area of Fe had a very low value of 1168, but the peak area of oxalic had a very high value of 5538. In addition, the peak area of iron exhibited a high value of 138199 at 14:00, while the peak area of oxalic acid showed a lower peak of 7687 at 14:00 and a higher peak of 11879 at 19:00 with an extreme low abundance of iron. Above asynchronous variation of iron and oxalic acid in iron rich HM type particles during the photochemical activity period from 5:00 to 19:00 strongly indicated that photolysis of iron oxalato complexes could be an efficient sink of oxalic acid." in lines 374-393.

"The HM type particles was the most abundant oxalic acid particles in summer and the diurnal variations of peak area of iron and oxalic acid show opposite trends, which suggest a possible loss of oxalic acid through the photolysis of iron oxalato complexes during the strong photochemical activity period." has been added in the abstract in lines 55-58.

"The diurnal variations of peak area of iron and oxalic acid in HM type particles indicate a net production of oxalic acid at 15:00 lower than at 19:00, likely due to a significant loss of oxalic acid through the photolysis of iron oxalato complexes during the strong photochemical activity period." has been added in the conclusion in lines 480-483.

[Figure]

Figure 8. The diurnal variations of peak area of iron (m/z=56) and oxalic acid (m/z=-89) in the HM type oxalic acid particles from July 28 to August 1, 2014.

3) Acid catalysis

By definition, a catalysis is a process where a compound (a catalyst) is involved in the reaction but will be recycled and not be consumed.

The authors refer to 'acid-catalysed oxalate formation from glyoxal' (l. 447). These studies did not show acid catalyzed reactions.

**Response**: In the abstract "which suggests the formation of oxalic acid is closely associated with acid-catalyzed reactions of organic precursors." has been changed to "which suggests the formation of oxalic acid is closely associated with acidic aqueous phase chemical processing of organic precursors." in lines 65-66.

"Several studies have reported potential production of oxalic acid from acid-catalyzed aqueous phase reactions in aerosols" has been changed to "Several studies have reported potential production of oxalic acid from acidic aqueous phase reactions in aerosols" in lines 422-423.

"which can be used as an indicator of acid-catalyzed ageing process of organic compounds." has been changed to "which can be used as a marker of acidic aqueous phase aging process of organic compounds" in lines 453-454.

"Based on the above discussion, the degradation of carbonaceous species associated with acid-catalyzed reactions may have an important contribution to the formation of oxalic acid during the episode in winter." has been changed to "Based on the above discussion, the degradation of carbonaceous species associated with acidic aqueous phase chemical reactions may have an important contribution to the formation of oxalic acid during the episode in winter." in lines 462-465.

"which suggest the acid-catalyzed oxidation of organic precursors as a potential source for oxalic acid." has been changed to "which suggests the acidic aqueous phase chemical processing of organic precursors as a potential source for oxalic acid." in lines 467-468.

"Nitric acid and organosulfate were found to co-exist in oxalic acid-containing particles in the winter, which suggests a close association with acid-catalyzed reactions. Acid-catalyzed oxidation of organic precursors is a potential contribution for the formation of oxalic acid in winter." has been changed to "Nitric acid and organosulfate were found to co-exist in oxalic acid-containing particles in the winter, which suggests a close association with acidic aqueous phase reactions. Acidic aqueous phase chemical processing of organic precursors is a potential contribution for the formation of oxalic acid in winter." in lines 486-489.

4) Trivialities

At several places, the authors include sentences that do not add to the discussion but are circular and trivial. Examples include

- l. 302: The low percentage of NH4+ in oxalic acid particles in summer indicated the presence of oxalic acid in NH4+ poor particles.

… do you want to say here that oxalic acid is rather accumulated in particles that do not contain NH4+?

- l. 438: Nitrate, sulfate and oxalic acid showed very similar variation patterns in winter, indicating a close connection of their coexistence.

… what is meant by 'connection of coexistence'? – it seems redundant.

**Response**: "However, the percentage of $NH_4^+$ with oxalic acid was only 18% in summer but increased to 71% in winter. Linear correlations between $NH_4^+$-containing oxalic acid particles ($C_2$-$NH_4^+$) and total oxalic acid particles are depicted in Figure 3, with better linear regression ($r^2$=0.98) in winter than summer. The low percentage of $NH_4^+$ in oxalic acid particles in summer indicated the presence of oxalic acid in $NH_4^+$-poor particles." has been changed to "However, the $NH_4^+$-containing oxalic acid particle ($C_2$-$NH_4^+$) only accounted for 18% of total oxalic acid particles in summer but this fraction increased to 71% in winter, and linear correlation between $C_2$-$NH_4^+$ particles and total oxalic acid particles showed better linear regression ($r^2$=0.98) in winter than summer, indicating a general mixing state of $NH_4^+$ with oxalic acid in winter." in lines 300-304.

"Nitrate, sulfate and oxalic acid showed very similar variation patterns in winter, indicating a close connection of their co-existence." has been changed to "Nitrate, sulfate and oxalic acid showed very similar variation patterns in winter, suggesting a close connection of the formation of oxalic acid with the existence of nitrate and sulfate." in lines 412-414.

Besides, we have examined the whole manuscript and revised the following sentence due to same problem. "The prominent photochemical feature of oxalic acid particles suggested a close association of photochemical reactions with oxalic acid production." has been removed in line 340.

**Minor comments**

1. line 60: 'relative acidity ratio' should be defined here (or simply called 'acidity')

**Response**: "The general existence of nitric acid and high relative acidity ratio in oxalic acid-containing particles indicates an acidic environment during the formation process of oxalic acid." has been changed to "The strong acidity and general existence of nitric acid in oxalic acid-containing particles indicates an acidic environment during the formation process of oxalic acid." in lines 60-62.

2. line 111-112: The mechanism of oxalic acid as described in the manuscript is not uncertain. The main uncertainty are the oxidant levels - as correctly stated in the response to the reviews, but is should be added here.

**Response**: "However, the detailed formation mechanisms of oxalic acid from photochemistry and aqueous phase chemistry in ambient aerosols are still not comprehensively understood and need to be further studied." has been changed to "However, the detailed formation mechanisms of oxalic acid from photochemistry and aqueous phase chemistry in ambient aerosols are still not comprehensively

understood due to the great uncertainty of oxidant levels, and need to be further studied." in lines 111-114.

3. line 369-371: This is completely vague and does not have any observational basis. This should be accompanied by a more detailed analysis.

**Response**: We have removed these sentences in the manuscript due to the uncertainty of the role of Fenton reactions in the formation process of oxalic acid. Please see the detailed discussion and revisions in the major comment 1.

4. line 378: The figure merely shows RH and some inorganic compounds etc. The figure does not 'discuss' anything but only shows time traces.
In addition, I suggest showing the H+ concentration in this figures as mol/L(aq) so that it can help the discussion of possible acidity effects.

**Response**: The concentration of $H^+$ has been changed from nmol/m to mol/L in Figure S5.

[Figure]

Figure S5. The diurnal variations of in-situ pH ($pH_{is}$), RH, nitrate, sulfate, ammonium and the aqueous phase concentration of $H^+$ (mol $L^{-1}$) in aerosols from July 28 to August 1 in 2014.
    "LWC is strongly dependent on the ambient RH and affected by water-soluble composition in the aerosols. Thus we investigate the diurnal patterns of RH, major inorganic ions and the free amount of $H^+$ in Figure S5. Although RH increased from 00:00 to 07:00, $H^+$ had higher concentration during this period than the other time, which resulted the lower value of $pH_{is}$ (between -1.42 and 0) from 00:00 to 05:00. Thus more acidic aerosols with $pH_{is}$ below 0 were observed due to the combined effects of the RH and relative abundance of $H^+$ in aerosols." has been changed to "LWC is strongly dependent on the ambient RH and water-soluble inorganic salts like sulfate, nitrate and ammonium in the aerosols. The aqueous phase concentration of $H^+$ was lower from 12:00 to 21:00 compared to other time, which suggests a less acidity

effect on the photochemical production of oxalic acid during this period." under the Figure S5.

5. line 394: replace 'due to reported studies' by 'according to previous studies' (or something similar)

**Response**: "the precursors of oxalic acid such as glyoxylic acid have higher reaction rate with •OH in high pH solutions due to reported studies" has been changed to "the precursors of oxalic acid such as glyoxylic acid have higher reaction rate with •OH in high pH solutions according to previous studies" in lines 354-356.

6. line 414: During haze periods, photochemical activity is usually reduced.

**Response**: Because we have removed the discussion of the role of Fenton reactions in the formation process of oxalic acid, the implication of metals in the formation of SOA has also been deleted in the manuscript. "A large amount of Fe related particles are emitted from steel industries in the North China Plain and metals like V, Zn, Cu and Pb from electronic manufacturing (Cui and Zhang, 2008;Dall'Osto et al., 2008). These metals contribute significantly to haze episodes (Moffet et al., 2008;Li et al., 2014), which possibly increases the formation of SOA by yielding more •OH participating the heterogeneous and aqueous reactions." has been removed in line 370.

7. line 429: Remove 'particles' here. 'Oxalic acid particles are prevalent in carbonaceous particles' sounds awkward.

**Response**: "oxalic acid particles were still prevalent in carbonaceous particles" has been changed to "oxalic acid was still prevalent in carbonaceous particles" in lines 403-404.

**Below are the responses to the reviewer's 2$^{nd}$ round comments.**

1. (a) Reviewer, 2$^{nd}$ round: The Fenton reaction is NOT a photolysis of $H_2O_2$. A photolysis would yield two OH radicals, whereas Fenton reaction leads to OH + OH$^-$

**Response**: We are sorry for this mistake. The discussion of Fenton reaction has been removed from the manuscript. "the photolysis of $H_2O_2$ through Fenton reactions involving the catalysis of transition metal ions like $Fe^{2+/3+}$, $Cu^{+/2+}$ and $Mn^{2+/3+}$ likely contributes substantially to the source of •OH in the aqueous phase in this work" has been removed.

(b) Reviewer, 2$^{nd}$ round: l. 383-392: While the authors add some text about the significant loss of oxalic acid due to the photolysis of the iron-oxalato-complex, this discussion is neither balanced nor well connected with the rest of the discussion. If iron is indeed abundantly available for Fenton reaction, why doesn't it show an impact on oxalic acid during day time when – according to literature – up to 99% of oxalic acid might be reduced?

**Response**: The sink of oxalic acid via the photolysis of iron oxalato complexes has been discussed through the diurnal variation of peak area of iron and oxalic acid in HM type particles. Please see the major comment 2.

(c) Reviewer, 2$^{nd}$ round: The same is true for the loss of oxalic acid by OH reactions. This should be mentioned here.

**Response**: It is true that the higher pH is favorable for the dissociation of oxalic acid to oxalate, however, the kinetic coefficient for oxalate with OH reaction is $(1.9\pm0.6)\times10^8$ at pH=3 and $(1.6\pm0.6)\times10^8$ at pH=8 according to Herrmann's work (Herrmann, 2003). Thus it is difficult to say the higher pH is favorable for the loss of oxalic acid via •OH reactions without further quantitative result.

(d) Reviewer, 2$^{nd}$ round: At low pH, when oxalic acid is not dissociated, the Henry's law constant is 9000 M/atm. That means only a very small fraction (<<1%) of all oxalic acid is expected to be in the aqueous phase of particles. Therefore, the authors' arguments here are not convincing.

**Response**: "Oxalate has a very low pKa (1.23) so that even at low pH a substantial fraction is still present as oxalate." has been mentioned by the referee in the 1$^{st}$ round comment (c), and in this work the $pH_{is}$ of ambient particles ranged from 1.5 to 4.01 from 12:00 to 21:00, which is suitable for the oxalic acid to dissociate to oxalate.

(e) Reviewer, 2$^{nd}$ round: In this figure, the loss of oxalate due to the photolysis of the iron-oxalato complex is missing.

**Response**: Thanks for your suggestion, and we have removed this figure and related discussion from the manuscript and supplement.

2. Reviewer, 2$^{nd}$ round: Thanks for clarifying.

**Response**: Thank you for your comment.

3. Reviewer, 2$^{nd}$ round: Given all these uncertainties and the fact that the authors do

not have any measurements on iron availability, I suggest to word the role of Fenton chemistry much more carefully throughout the whole manuscript.

**Response**: According to the referee's 1$^{st}$ and 2$^{nd}$ round comments, we think that more field measurements and data are needed to comprehensively and qualitatively discuss the contribution of Fenton reaction source •OH radicals to oxalic acid formation. This is out of the interest of this work. Thus we prefer to remove the discussion of the contribution of Fenton reaction in the formation process of oxalic acid and the proposed mechanism along the suggestion of the referee. Fenton reaction is only mentioned as a possibility according to the literature in a very cautious expression. Please see the revisions and changed in major comment 1.

4. Reviewer, 2nd round: I appreciate this change.

**Response**: Thank you for your comment.

5. Reviewer, 2nd round: The studies by Tan et al., Lim et al. and other by the Turpin group and others, have shown that oxalic acid is not efficiently formed in aqueous aerosol particles. – as I had pointed to in my previous comment.

Either the authors need to clarify that despite of RH < 100% (i.e. no clouds) at the study location, the observed oxalic acid was likely formed in clouds prior to arriving at the study location (what data is available to prove cloudiness) or they should at least briefly mention the more significant formation of oligomers as shown in the cited references.

**Response**: "The aqueous phase oxidation of glyoxal can take place in both of clouds and wet aerosols (Lim et al., 2010). However, the lower yield of oxalic acid from glyoxal in wet aerosols compared to in clouds has been reported in previous chamber experiments due to the formation of substantial amount of high molecular weight products such as oligomers in aerosols-relevant concentrations (Carlton et al., 2007;Tan et al., 2009). These findings may explain the lower peak of oxalic acid particles at 15:00 compared to that at 19:00." has been added in the manuscript in lines 348-354.

6. Reviewer, 2nd round: Why is higher temperature favorable for more SOA formation? Rate constants usually increase. However, they do so for both formation rates and loss rates of oxalic acid. In addition, higher temperature also favor higher evaporation of volatile gases, including water.

**Response**: We believe this expression "The high temperature between 9:00 and 19:00 was favorable to the secondary processing of organic precursors." is vague and ambiguous. Since the diurnal pattern of temperature is routine and the mixing state of oxalic acid with major chemical species has been discussed in the manuscript, we decide to remove this sentence.

"The diurnal patterns of temperature, wind speed are presented in Figure S7. The high temperature between 9:00 and 19:00 was favorable to the secondary processing of organic precursors. The wind speed was low during the whole day, especially between 9:00 and 18:00, which provided a stagnant environment for the increase in oxalic acid produced from photochemical process." has been changed to "The wind speed was low during the whole day (Figure S6), especially between 9:00 and 18:00, which provided a stagnant environment for the increase in oxalic acid produced from photochemical process." in lines 399-401.

7. Reviewer, 2nd round: Thanks for clarification.
**Response**: Thank you for your comment.

**References:**

[revised manuscript text omitted]

---

## Author Response (AR3)

**Response to the comments**

[Atmospheric Chemistry and Physics, MS ID: acp-2016-1081]
Title: Mixing state of oxalic acid containing particles in the rural area of Pearl River Delta, China: implications for the formation mechanism of oxalic acid

**Comments to the Author:**

After the corrections/changes based on the Reviewer's comments (especially, main comment), the paper could be ready for acceptance.

**Response**: Many thanks for your reviewing and comments. We have revised our manuscript according to the reviewer's comments, and our point by point responses to the comments have been shown in the following section. We appreciate your comments and response that enable current study to meet the high quality of the journal *Atmos. Chem. Phys.*.

Anything about our paper, please feel free to contact me at limei2007@163.com

Best regards!

Sincerely yours
Mei Li
July 6, 2017

**Specific comments and point by point responses:**

**Main comments**

  Throughout the manuscript, the authors state that 'acidic aqueous phase chemical processing' leads to oxalic acid. I still do not understand why the authors keep emphasizing the role of acidity. In general, the rate constants of dicarboxylates (e.g. glyoxylate, pyruvate) decrease with acidity. Or are the authors saying that the LOSS PROCESSES of oxalic acid are slowed down at higher acidity and therefore oxalic acid accumulated? If so, that should be mentioned somewhere that the pH was sufficiently low that indeed oxalic acid ($H_2C_2O_4$) and not its anions ($HC_2O_4^-$, $C_2O_4^{2-}$) was predominant. However, this only occurs at extremely low pH, i.e. if pH ~ $pKa(H_2C_2O_4)$ ~ 1.25.

**Response**: According to reported studies in field measurements, chamber experiments and model simulations(Carlton et al., 2007;Tan et al., 2009;Myriokefalitakis et al., 2011;Wang et al., 2012;Kawamura et al., 2013), oxalic acid is one of the main products from the oxidation of organic precursors like glyoxal in the aqueous phase. Although the influence of different particle acidity on the oxidation process of glyoxal still needs evaluation, the moderate acidic environment is favorable for the production of oxalic acid from the oxidation of glyoxal (Herrmann, 2003;Ervens and Volkamer, 2010;Eugene et al., 2016). In this work the general existence of nitric acid and the temporal variation of relative acidity ratio (from 7 to 114) both indicated an acidic environment of oxalic acid particles during the sampling period in winter. 98% of oxalic acid particles contained sulfate suggesting a strong connection between the aqueous phase reactions and the formation process of oxalic acid. The abundant hydrocarbon fragments and secondary ions in the mass spectra of oxalic acid particles indicated the aging process of organic precursors in oxalic acid particles. These observations suggest a possible production of oxalic acid from the oxidation of glyoxal and related precursors in the acidic aqueous phase in aerosols. Besides, the formation of organosulfate from the reaction of glyoxal and sulfuric acid requires an acidic aqueous phase environment. The robust correlation between oxalic acid and organosulfate particles in the episode suggests the formation of oxalic acid is closely associated with the oxidation of organic precursors in the acidic aqueous phase. Based on the above discussion, we believe the acidity of oxalic acid particles plays an important role in the oxidation process of organic precursors in aqueous phase.

  Because the in-situ pH of oxalic acid particles was not measured in winter, and oxalic acid with its anion (oxalate) could not be characterized by the SPAMS, thus the loss process of oxalic acid through the oxidation of its anion were not discussed in winter. Several revisions have been made in the manuscript to describe the formation process of oxalic acid more clearly:

  In the abstract "which suggests the formation of oxalic acid is closely associated with acidic aqueous phase chemical processing of organic precursors" has been revised to "which suggests the formation of oxalic acid is closely associated with the oxidation of organic precursors in aqueous phase" in lines 65-66.

"Several studies have reported potential production of oxalic acid from acidic aqueous phase reactions in aerosols" has been revised to "Several studies have reported the formation of oxalic acid through the oxidation of glyoxal and related precursors in acidic aqueous phase" in lines 422-423.

"Although the influence of different particle acidity on the oxidation process of glyoxal still needs evaluation, the moderate acidic environment is favorable for the production of oxalic acid from the oxidation of glyoxal (Herrmann, 2003;Ervens and Volkamer, 2010;Eugene et al., 2016)." has been added in lines 424-428.

**Minor corrections and revisions**

1. line 55: replace 'was' by 'were'

**Response**: "The HM type particles was the most abundant oxalic acid particles in summer and the diurnal variations of peak area of iron and oxalic acid show opposite trends" has been revised to "The HM type particles were the most abundant oxalic acid particles in summer and the diurnal variations of peak area of iron and oxalic acid show opposite trends" in lines 55-56.

2. line 60 – 62: This sentence seems meaningless and redundant "The strong acidity… indicated an acidic environment…"

**Response**: "The strong acidity and general existence of nitric acid in oxalic acid-containing particles indicates an acidic environment during the formation process of oxalic acid." has been revised to "The general existence of nitric acid in oxalic acid-containing particles indicates an acidic environment during the formation process of oxalic acid." in lines 60-62.

3. line 111: The formation MECHANISM of oxalic acid is realtively well constrained, i.e. the facts that glyoxal is oxidized to glyoxylic acid and further to oxalate. However, the oxidation RATES are associated with uncertainties as they are the product of the reactants (organics and oxidants) and the rate constants.

**Response**: "However, the detailed formation mechanisms of oxalic acid from photochemistry and aqueous phase chemistry in ambient aerosols are still not comprehensively understood due to the great uncertainty of oxidant levels, and need to be further studied." has been revised to "However, the formation process of oxalic acid in ambient aerosols is still associated with great uncertainty due to the oxidation rates of precursors and oxidant levels in photochemistry and aqueous phase chemistry, which needs to be further studied." in lines 111-114.

4. line 342: As mentioned in my previous comments, the figure does not DISCUSS the influence of the pH. A discussion would include a line of arguments on the various processes that include RH, inorganic ions etc and how and to what extent they affect pH. Since this information cannot be gathrered form Fig. S5, it is not a discussion but merely the representation of temporal trends of pH, RH, inorganic ions etc.

**Response**: "The $pH_{is}$ of ambient particles ranging from -1.42 to 4.01 indicated an acidic environment, and the influences of $pH_{is}$ from RH, inorganic ions and $H^+$ (aq) in aerosols are discussed in Figure S5." has been revised to "The $pH_{is}$ of ambient particles ranging from -1.42 to 4.01 indicated an acidic environment, and the temporal trends of RH, inorganic ions and $H^+$ (aq) in aerosols are shown in Figure S5." in lines 340-342.

5. line 349: Remove 'of'

**Response**: "in both of clouds and wet aerosols" has been revised to "in both clouds and wet aerosols" in line 349.

6. line 352: replace 'aerosols-related' by 'aerosol-related'

**Response**: "oligomers in aerosols-relevant concentrations" has been revised to "oligomers in aerosol-related concentrations" in line 352.

7. line 383: Why 'likely correlated'? There are robust statistical methods that can be applied to confirm (or refuse) a statistical correlation.

**Response**: "Interestingly, the peak area of iron likely anti-correlated with the peak area of oxalic acid from 4:00 to 11:00." has been revised to "Interestingly, the peak area of iron exhibited opposite trend with the peak area of oxalic acid from 4:00 to 11:00." in lines 383-384.

8. line 389: replace 'extreme' by 'extremely'

**Response**: "an extreme low abundance of iron" has been revised to "an extremely low abundance of iron" in lines 389-390.

9. line 390: What do you mean by 'above asynchronous variation'? I suggest replacing it by a more common expression.

**Response**: "Above asynchronous variation of iron and oxalic acid in iron rich HM type particles during the photochemical activity period from 5:00 to 19:00 strongly indicated that photolysis of iron oxalato complexes could be an efficient sink of oxalic acid." has been revised to "Above opposite variation patterns of iron and oxalic acid in iron rich HM type particles during the photochemical activity period from 5:00 to 19:00 strongly indicated that photolysis of iron oxalato complexes could be an efficient sink of oxalic acid." in lines 390-393.

References:

Carlton, A. G., Turpin, B. J., Altieri, K. E., Seitzinger, S., Reff, A., Lim, H. J., and Ervens, B.: Atmospheric oxalic acid and SOA production from glyoxal: Results of aqueous photooxidation experiments, Atmospheric Environment, 41, 7588-7602, 10.1016/j.atmosenv.2007.05.035, 2007.

Ervens, B., and Volkamer, R.: Glyoxal processing by aerosol multiphase chemistry: towards a kinetic modeling framework of secondary organic aerosol formation in aqueous particles, Atmospheric Chemistry and Physics, 10, 8219-8244, DOI 10.5194/acp-10-8219-2010, 2010.

Eugene, A. J., Xia, S. S., and Guzman, M. I.: Aqueous Photochemistry of Glyoxylic Acid, Journal of Physical Chemistry A, 120, 3817-3826, 10.1021/acs.jpca.6b00225, 2016.

Herrmann, H.: Kinetics of aqueous phase reactions relevant for atmospheric chemistry, Chem Rev, 103, 4691-4716, Doi 10.1021/Cr020658q, 2003.

Kawamura, K., Tachibana, E., Okuzawa, K., Aggarwal, S. G., Kanaya, Y., and Wang, Z. F.: High abundances of water-soluble dicarboxylic acids, ketocarboxylic acids and alpha-dicarbonyls in the mountaintop aerosols over the North China Plain during wheat burning season, Atmospheric Chemistry and Physics, 13, 8285-8302, 10.5194/acp-13-8285-2013, 2013.

Myriokefalitakis, S., Tsigaridis, K., Mihalopoulos, N., Sciare, J., Nenes, A., Kawamura, K., Segers, A., and Kanakidou, M.: In-cloud oxalate formation in the global troposphere: a 3-D modeling study, Atmospheric Chemistry and Physics, 11, 5761-5782, 10.5194/acp-11-5761-2011, 2011.

Tan, Y., Perri, M. J., Seitzinger, S. P., and Turpin, B. J.: Effects of Precursor Concentration and Acidic Sulfate in Aqueous Glyoxal-OH Radical Oxidation and Implications for Secondary Organic Aerosol, Environmental Science & Technology, 43, 8105-8112, 10.1021/Es901742f, 2009.

Wang, G. H., Kawamura, K., Cheng, C. L., Li, J. J., Cao, J. J., Zhang, R. J., Zhang, T., Liu, S. X., and Zhao, Z. Z.: Molecular Distribution and Stable Carbon Isotopic Composition of Dicarboxylic Acids, Ketocarboxylic Acids, and alpha-Dicarbonyls in Size-Resolved Atmospheric Particles From Xi'an City, China, Environmental Science & Technology, 46, 4783-4791, 10.1021/es204322c, 2012.